# LMO7 deficiency reveals the significance of the cuticular plate for hearing function

Ting-Ting Du[1], James B. Dewey[2], Elizabeth L. Wagner[1], Runjia Cui[3], Jinho Heo[4], Jeong-Jin Park[5], Shimon P. Francis[1], Edward Perez-Reyes[6], Stacey J. Guillot[7], Nicholas E. Sherman[5], Wenhao Xu[8], John S Oghalai[2], Bechara Kachar[3] & Jung-Bum Shin[1]

Sensory hair cells, the mechanoreceptors of the auditory and vestibular systems, harbor two specialized elaborations of the apical surface, the hair bundle and the cuticular plate. In contrast to the extensively studied mechanosensory hair bundle, the cuticular plate is not as well understood. It is believed to provide a rigid foundation for stereocilia motion, but specifics about its function, especially the significance of its integrity for long-term maintenance of hair cell mechanotransduction, are not known. We discovered that a hair cell protein called LIM only protein 7 (LMO7) is specifically localized in the cuticular plate and the cell junction. *Lmo7 KO* mice suffer multiple cuticular plate deficiencies, including reduced filamentous actin density and abnormal stereociliar rootlets. In addition to the cuticular plate defects, older *Lmo7 KO* mice develop abnormalities in inner hair cell stereocilia. Together, these defects affect cochlear tuning and sensitivity and give rise to late-onset progressive hearing loss.

[1] Department of Neuroscience, University of Virginia, Charlottesville, VA 22908, USA. [2] Caruso Department of Otolaryngology-Head and Neck Surgery, University of Southern California, Los Angeles, CA 90033, USA. [3] National Institute for Deafness and Communications Disorders, National Institute of Health, Bethesda, MD 20892, USA. [4] Center for Cell Signaling and Department of Microbiology, Immunology and Cancer Biology, University of Virginia, Charlottesville, VA 22908, USA. [5] Biomolecular Analysis Facility, University of Virginia, Charlottesville, VA 22908, USA. [6] Department of Pharmacology, University of Virginia, Charlottesville, VA 22908, USA. [7] Advanced Microscopy core, University of Virginia, Charlottesville, VA 22908, USA. [8] Genetically Engineered Murine Model (GEMM) core, University of Virginia, Charlottesville, VA 22908, USA. Correspondence and requests for materials should be addressed to J.-B.S. (email: js2ee@virginia.edu)

In few other cell types is the principle of Form follows Function as evident as in the sensory hair cell. The hair cell's subcellular structures are optimally designed to facilitate hair cell mechanotransduction, the process by which mechanical energy from sound and head movements are converted into cellular receptor potentials. Two specialized structures at the apical surface of the hair cell, the hair bundle, and the cuticular plate, are essential for hair cell mechanotransduction[1–7]. Both are hair cell-specific elaborations of structures found in other microvilli-bearing cells, such as intestinal brush border cells[8,9]. The hair bundle, an array of microvilli arranged in a staircase-like fashion, harbors the mechanotransduction complex. A substantial body of research has identified the mechanisms essential for the morphogenesis and function of the hair bundle[10–16]. In contrast, the molecular composition and significance of the cuticular plate, a structure analogous to the brush border cell terminal web, is poorly understood (Fig. 1a). The cuticular plate is believed to provide a mechanical foundation for the stereocilia, which are inserted into it[17–20]. A stiff stereociliar insertion point ensures that vibration energy is fully converted into stereocilia pivot motion, and not diminished by non-productive cuticular plate deformations. This notion is supported by electron microscopy-based ultrastructural studies, which demonstrate that the cuticular plate is reinforced by a dense network of actin filaments, crosslinked by actin-binding proteins such as spectrin[21]. In addition to providing a mechanical foundation, the cuticular plate is also believed to be involved in selective apical trafficking of proteins and vesicles[22]. However, specifics about the function and formation of the cuticular plate, especially the significance of its integrity for long-term maintenance of hair cell function, are unknown. This gap in knowledge is in part attributable to the lack of molecular tools to manipulate the cuticular plate specifically. Molecular studies have uncovered a few resident proteins, such as spectrin, tropomyosin, supervillin[23–27], but loss-of-function studies for these proteins have not been undertaken to date.

In this study, we report the discovery of a novel component of the cuticular plate, a hair cell-enriched protein called LIM only protein 7 (LMO7). LMO7 contains a calponin homology (CH) domain, a PDZ domain, and a LIM domain, and was reported to be involved in protein-protein interactions at adherens junctions and focal adhesions[28,29]. LMO7 deficiencies have been reported to increase susceptibility to spontaneous lung cancer and contribute to the pathology of Emery-Dreifuss Muscular Dystrophy (EDMD)[30,31], although the latter hypothesis is controversial[32]. In addition, LMO7 was revealed to play a role in the regulation of actin dynamics through the Rho-dependent MRTF-SRF signaling pathway[33]. Our studies show that in the hair cell, LMO7 is specifically localized to the cuticular plate and intercellular junctions, where it plays a role in F-actin network organization. In LMO7-deficient mice, hair cells exhibit reduced F-actin staining in the cuticular plate, along with abnormal distribution of stereocilia rootlets. In addition, older *Lmo7 KO* mice develop abnormalities in the stereocilia of inner hair cells. These morphological defects affect tuning and sensitivity of cochlear vibrations, and cause late-onset progressive hearing loss.

## Results

### LMO7 is an abundant component of the cuticular plate.
We previously used a peptide mass spectrometry-based strategy to characterize the hair bundle proteome of chick vestibular organs, and identified LIM-only protein 7 (LMO7) as a potential hair bundle enriched protein[34,35] (Fig. 1b). Subsequent immunolocalization analysis showed that LMO7 is expressed in the mouse inner ear and highly enriched in the sensory hair cells (Fig. 1d, e and Supplementary Fig. 1a). Contrary to our expectations,

however, its localization was restricted to the cuticular plate and the intercellular junctions (Fig. 1f, g). The presence of LMO7 in the hair bundle preparation was likely due to co-purification of cuticular plate material during the hair bundle twist-off purification. This is evident when hair bundle preparations from mouse utricles embedded in the agarose matrix used for the twist-off method are immunolabeled with an LMO7-specific antibody (Fig. 1c). LMO7 immunoreactivity is abundant in the cuticular plate, but not in the hair bundle.

We next examined the spatiotemporal expression of LMO7 in the mouse inner ear using immunohistochemistry at various stages of development. Initial LMO7 expression in the embryonic cochlea coincides with the emergence of hair bundles at embryonic day 16 (E16), and its expression increases and continues into mature stages (Fig. 1d). Likewise, LMO7 is abundant during development of the utricle cuticular plate (Fig. 1e). Overall, the spatiotemporal expression of LMO7 is similar to spectrin, an actin-binding protein localized in the cuticular plate (Fig. 1d, e). In summary, LMO7 is a hair cell-enriched protein, with specific localization to the cuticular plate and the cell junctions.

LMO7 was reported to localize at various subcellular sites in other cell types. While most studies agree that LMO7 is localized at adherens junctions and focal adhesions, it was also reported that LMO7 might act as a nucleocytoplasmic shuttling protein to regulate transcription of emerin[36]. In order to ascertain the localization of endogenous LMO7 in hair cells, we employed an alternative method of protein localization using split-GFP (Fig. 2a). GFP features 11 beta strands making up a beta-barrel. Removing one of those beta-strands (creating the so-called GFP1-10) abolishes its fluorescence, but exogenous addition of the missing beta-strand (GFP11), which can be fused to a protein of interest, leads to spontaneous re-assembly and reconstitution of fluorescence. This strategy allows tagging and visualization of the protein of interest based on the localization of reconstituted GFP fluorescence. Split fluorescent protein has been widely used for protein quantification, visualization of protein cellular localization, as well as single-molecule imaging and protein-protein interaction[37–43]. Using CRISPR/Cas genome editing, we knocked-in the GFP11-coding sequence (48 nucleotides) into the C-terminus of mouse *Lmo7*, immediately prior to the stop codon (resulting in the *Lmo7-GFP11 KI mice*) (Fig. 2b, c). We confirmed that the GFP11 tag does not affect the expression and localization of LMO7 (Fig. 2d). Explant cultures were established from *Lmo7-GFP11 KI* mice at P2, and to achieve complementation with GFP1-10, we transduced organ of Corti explants with AAV2/Anc80 virus carrying the GFP1-10 fragment coding sequence[44], driven by a CMV promoter (Fig. 2e). As illustrated in Fig. 2f, outer hair cells in *Lmo7-GFP11 KI*, which were transduced with GFP1-10, displayed robustly reconstituted GFP fluorescence at the cuticular plate level but not at the level of the cell body (Fig. 2f, g). Despite the fact that GFP1-10 was detected throughout the cytosol and the nucleus, we never detected any specific GFP fluorescence in the nucleus, suggesting that at least in hair cells, LMO7 is unlikely to regulate transcription by nucleocytoplasmic shuttling.

### *Lmo7 KO* mice have defects in the cuticular plate.
To investigate the function of LMO7 in the hair cell and in hearing function, we sought to analyze LMO7 loss-of-function mice. We initially obtained sperm from a mouse strain generated at Texas A&M using gene trap technology[45] (mouse ID: Lmo7Gt(IST10208D3) Tigm) and generated the mice using in vitro fertilization. In these mice (*Lmo7 gene trap*), a cassette with a transcription-terminating polyadenylation site is inserted after exon 1. In a previous study, this mouse line was used to show that LMO7 is

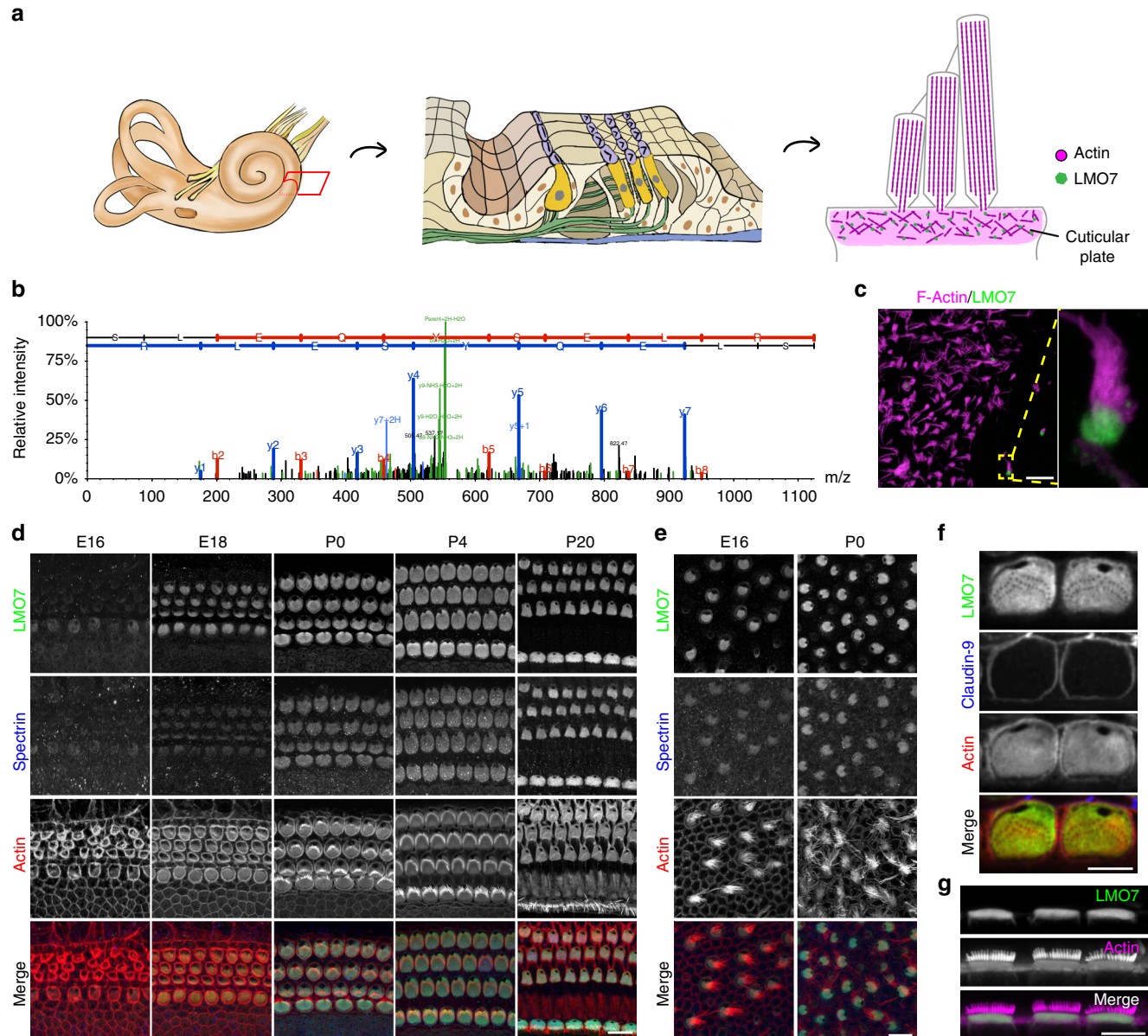

**Fig. 1** LMO7 is a component of the cuticular plate and the junctions. **a** Schematic representation of inner ear organization and the apical structures of the hair cell. **b** MS/MS spectrum of a representative peptide of chick LMO7, identified by LC-MS/MS on isolated chick hair bundles. The data for the spectrum was obtained from a previously published dataset[35]. **c** LMO7 immunoreactivity (green) in isolated mouse hair bundles confirmed its presence in cuticular plate (labeled by phalloidin in magenta). Scale bars, 20 μm (overview), 5 μm (panel magnification). **d**, **e** Immunohistochemical analysis of LMO7 expression in the mouse cochlea and utricle at various ages. LMO7 expression is detected in hair cells at E16, with the initial emergence of the hair bundle. Scale bar, 10 μm. **f** Higher-magnification views of LMO7 and claudin 9 immunoreactivity in mouse inner hair cells at the level of the cuticular plate. LMO7 localization is restricted to the cuticular plate and the intercellular junctions. **g** Side view of LMO7 expression in the inner hair cell. Scale bar, 5 μm

implicated in EDMD[31]. However, immunolabeling experiments showed that LMO7 protein expression in hair cells of *Lmo7 gene trap* mice was not abolished (Supplementary Fig. 1b). Immunoblot analysis further confirmed that the gene trap strategy only affected the longest isoform of LMO7 (Fig. 3d), consistent with the strategy to insert the gene trap cassette after exon 1. Our conclusion that the *Lmo7 gene trap* mouse does not represent a global *Lmo7 KO mouse* model is consistent with a previous study by Lao et al.[32]. We therefore used CRISPR/Cas9 to generate a variety of mice with potentially deleterious mutations in the *Lmo7* gene. LMO7 is expressed in multiple poorly characterized isoforms (Fig. 3a). In order to increase our chances of knocking out all isoforms, we targeted exon 12, exon 17 or exon 28 of the

mouse *Lmo7* gene. We isolated three founders, with a deleterious 2 bp insertion in exon 12 (*Lmo7 exon12 KO* mouse), a 5 bp deletion in exon 17 (*Lmo7 exon17 KO* mouse) or an 8 bp deletion in exon 28 (*Lmo7 exon28 KO* mouse), respectively (Fig. 3b). The founders were bred 4–5 generations to breed out potential off-target mutations, and homozygotes were analyzed. In the *Lmo7 exon17 KO* mice, in which the 5-bp deletion leads to a premature stop codon five amino acids downstream of the mutation site (Fig. 3b), LMO7 immunoreactivity was undetectable in auditory and vestibular hair cells (Fig. 3c). This was further confirmed by immunoblot analysis (Fig. 3d). The antibody used here (M-300, Santa Cruz) recognizes an epitope downstream of the deleterious mutation. Lack of signal in immunolabeling or blots therefore

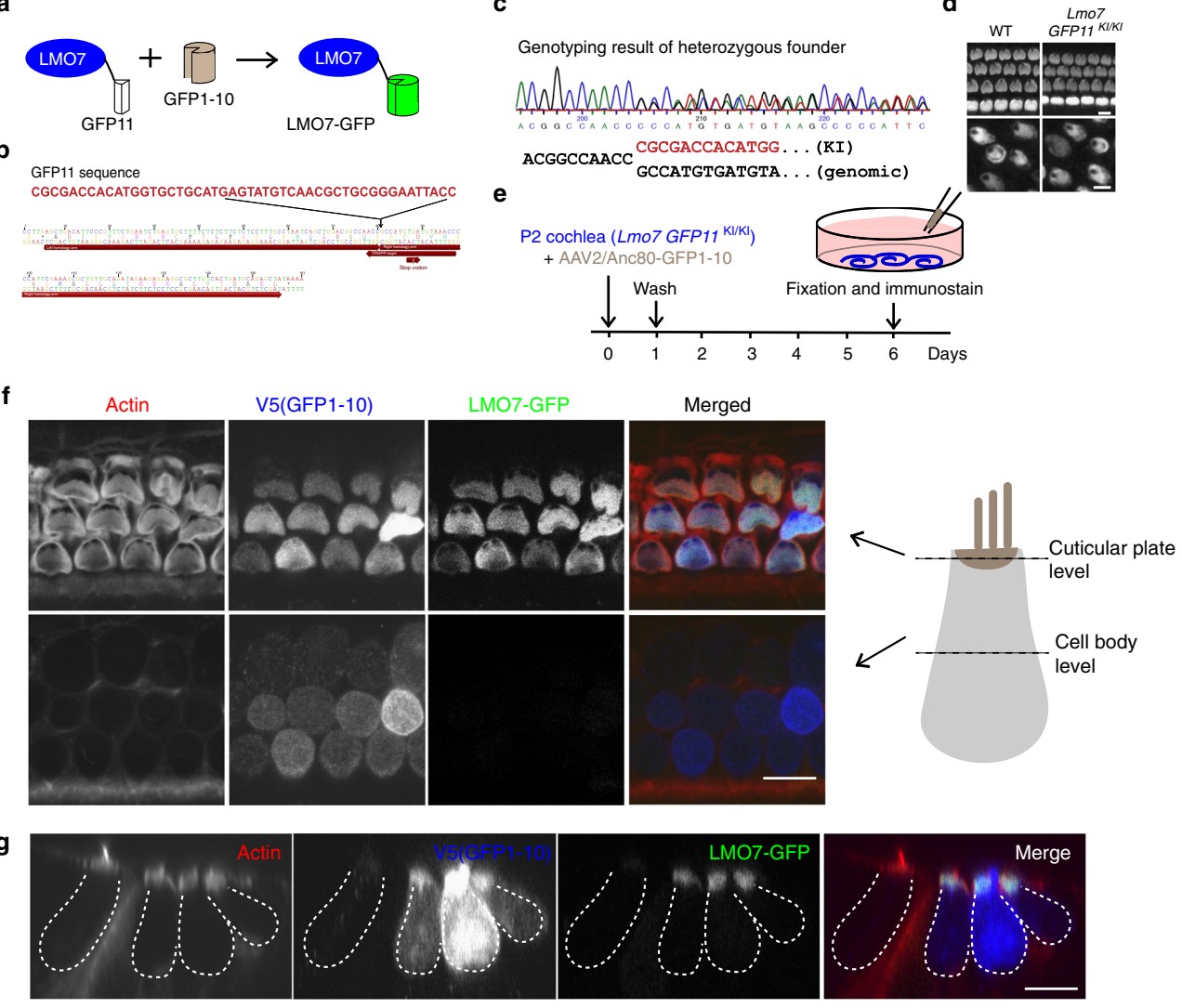

**Fig. 2** Localization of endogenous LMO7 using the split-GFP approach. **a** Schematic diagram of the split-GFP approach to label tagged endogenous LMO7. **b** Illustration of CRISPR/Cas9-mediated GFP11 knock-in strategy. Genomic sequence surrounding the stop codon of mouse *Lmo7*. Highlighted are the CRISPR target sites and the left and right homology arms used for the homology-directed repair. **c** Sequencing chromatogram of the modified locus, from the *Lmo7-GFP11 KI* founder mouse. **d** Immunofluorescence localization of LMO7 in P4 WT and *Lmo7-GFP11 KI/KI* cochlea and utricle. **e** Depiction of experimental paradigm for transduction of explants of *Lmo7-GFP11 KI* cochlea with AAV2/Anc80-GFP1-10. The cochleae were dissected and transduced with AAV2/Anc80-GFP1-10 for 24 h. After replacement of culture medium, cultures were maintained for 5 days, fixed, and immunostained. **f** Optical sections at the level of the cuticular plate and cell body are shown. Reconstituted LMO7-GFP expression localizes to the cuticular plate but not the cell body level. Scale bar, 5 µm. **g** Side view of reconstituted LMO7-GFP expression enriched in the cuticular plate of outer hair cell. Scale bar, 5 µm

does not exclude the possibility that a truncated variant is still expressed. We addressed this issue by confirming loss of immunolabeling and blot signals in *Lmo7 exon17 KO* tissue using an independent anti-LMO7 antibody (Sigma) that recognizes a region upstream of the mutation (indicated in Fig. 3a). This validates the *Lmo7 exon17 KO* mouse line as a bona fide loss-of-protein model.

The other mouse strains exhibited various degrees of incomplete ablation of LMO7 expression: *Lmo7 exon28 KO* mice showed reduced LMO7 immunoreactivity, and the LMO7 immunoreactivity is abolished in *Lmo7 exon12 KO* mice initially, but returns in the mature mice (supplementary Fig. 1b). We believe that this is caused by a change in splicing, induced by the mutation. This phenomenon, while interesting, is not expected to be important for understanding LMO7's function in hair cells. The remainder of the study thus focused on the *Lmo7 exon17 KO* mouse line.

*Lmo7 exon17 KO* mice are healthy, viable, and display no overt phenotype. The gross morphology of the sensory epithelium was not affected. Upon closer inspection, however, some striking differences were detected: The cuticular plate of inner and outer hair cells showed a significantly reduced amount of F-actin, as measured by phalloidin staining intensity (Fig. 4a, b). This pathological phenotype progressed in an age-dependent manner in outer hair cells but not inner hair cells (Fig. 4b). The degree of F-actin reduction was similar along the entire length of the organ of Corti and showed no tonotopic differences in inner or outer hair cells (Fig. 4c). Ultrastructural analysis using transmission electron microscopy (TEM) demonstrated that the cuticular plates in the mutant mice were markedly thinner compared to WT counterparts, in both IHCs and OHCs (Fig. 4d). Since LMO7 also localizes to cellular junctions, we investigated cell junction integrity. While TEM analysis did not indicate a gross defect of the junctions (Fig. 4d), immunofluorescence analysis revealed

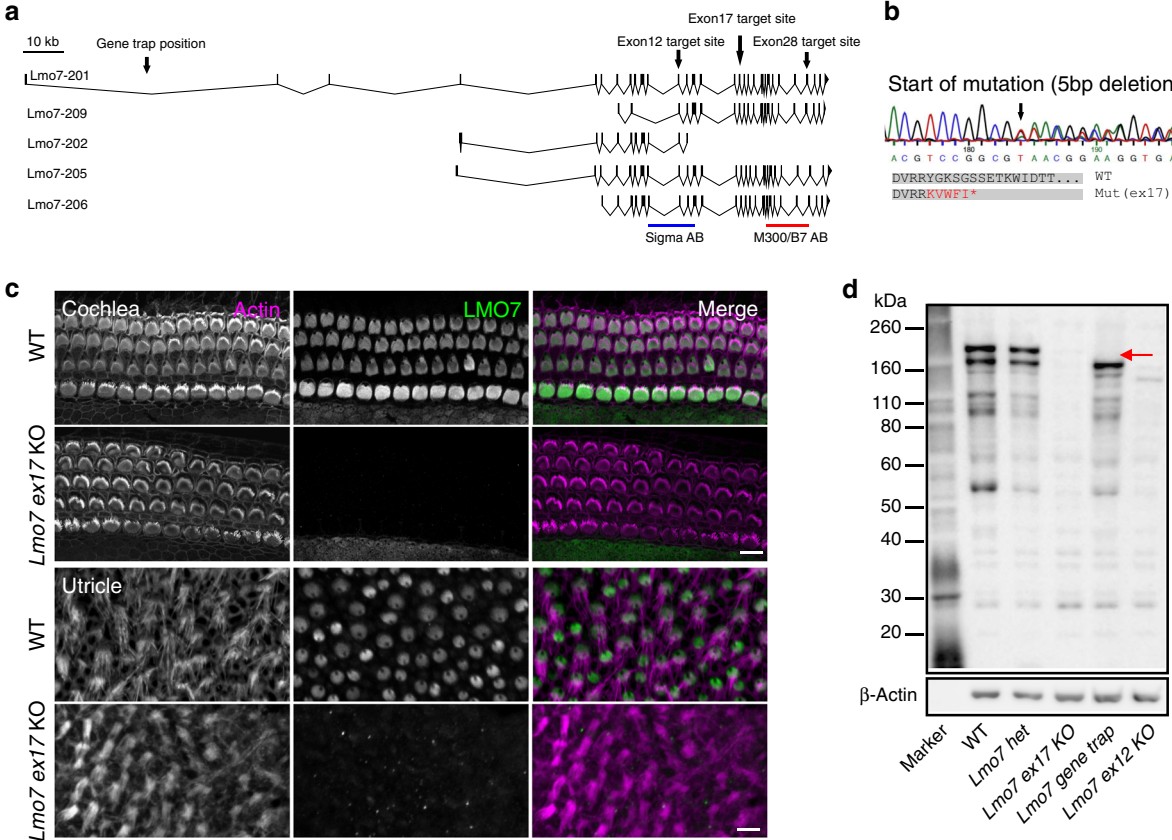

**Fig. 3** Generation of LMO7-deficient mice. **a** Genomic structures of mouse *Lmo7* isoforms from Ensembl (GRCm38.p5). Exons and 5′ untranslated regions are indicated by black boxes, 3′ untranslated regions are indicated by black triangles. The locations of the gene trap and three CRISPR-mediated targeting sites are shown with arrows. The epitope regions of antibodies used in this study are indicated in blue (Sigma AB) and red (Santa Cruz M-300 and B-7). **b** Sanger sequencing result of edited locus demonstrates successful genome editing, resulting in a 5 bp deletion in exon 17 which creates a premature stop codon as shown in grey shading. **c** LMO7 immunolabeling in WT and *Lmo7 exon17 KO* mice in P4 cochlea and utricle. LMO7 is in green, and F-actin in magenta. LMO7 immunoreactivity is abolished in KO cochlear and vestibular hair cells. **d** Comparative immunoblot of WT, *Lmo7 exon17 Het*, *Lmo7 exon17 KO*, *Lmo7 gene trap* and *Lmo7 exon12 KO* from P4 inner ear lysate, indicating loss of LMO7 expression in *Lmo7 exon17* lysate. The gene trap strategy only abolished the longest isoform, indicating by red arrow. The β-actin-specific bands indicate equal loading

that both F-actin and the tight junction component claudin 9 staining extended deeper into the cell body in WT mice compared to mutants (Fig. 4e and supplementary Fig. 2a). We next examined the rootlets that form the stereocilia insertion points into the cuticular plate. TRIOBP, an actin-bundling protein, is an essential component of the rootlet, and TRIOBP deficiency leads to sub-optimal mechanotransduction, causing hearing loss in mice and humans[46]. We found that the spatial organization of the rootlets was disrupted in *Lmo7 exon17 KO* mice (Fig. 4f, g). In addition, the length of TRIOBP-positive rootlets was significantly reduced in the *Lmo7 exon17 KO* mice, along with the intensity of TRIOBP staining (Fig. 4h). In summary, LMO7 deficiency compromises the F-actin network in the cuticular plate, and to a lesser degree, at the cell junctions, giving rise to thinner and less dense cuticular plates, and shortened cell junctions and stereocilia rootlets.

**LMO7 interacts with cuticular plate and junction components**. We next investigated the mechanism by which LMO7 affects the F-actin organization in the cuticular plate and the junctions. To identify inner ear-specific interaction partners, we carried out a co-immunoprecipitation from P6 mouse organ of Corti using an LMO7-specific antibody (mouse monoclonal B-7), followed by protein identification by peptide mass spectrometry (Co-IP/MS).

Co-IP from an equivalent amount of *Lmo7 KO* tissue was used as a negative control. Approximately 500 proteins were identified (the full data sets are available as Supplementary Data 1 (WT Co-IP) and Supplementary Data 2 (KO Co-IP). To focus on the most significant interaction partners, we selected the 40 most abundant proteins in the WT pulldown, and calculated the fold-changes of these proteins in the WT compared to the KO pulldown sample (Table 1). As expected, LMO7 was identified in the WT pulldown, as the third most abundant protein (97 spectral counts), and absent in the KO pulldown. Twenty-six out of the 40 proteins were more abundant in the WT pulldown (fold change >1), among them known cuticular plate proteins, such as spectrins (SPTBN1, 2.9-fold and SPTAN1, 2-fold), Myosin VI (MYO6) (5.8-fold) and actin isoforms (ACTB representing all actins, 1.2-fold). Known cell junction proteins such as subunits of non-muscle myosin II (NMII) (MYH14, detected in WT only and MYH9, 5.3-fold) and α-actinin 4 (ACTN4, 10-fold) were also enriched in the WT sample.

**LMO7 reorganizes the F-actin network**. Of the 25 proteins enriched in the WT sample, nine are known to associate with the F-actin cytoskeleton, including actins, myosins, spectrins, actinin, and serpin. We therefore tested the effect of LMO7 on F-actin organization in a heterologous cell line. COS-7 cells were

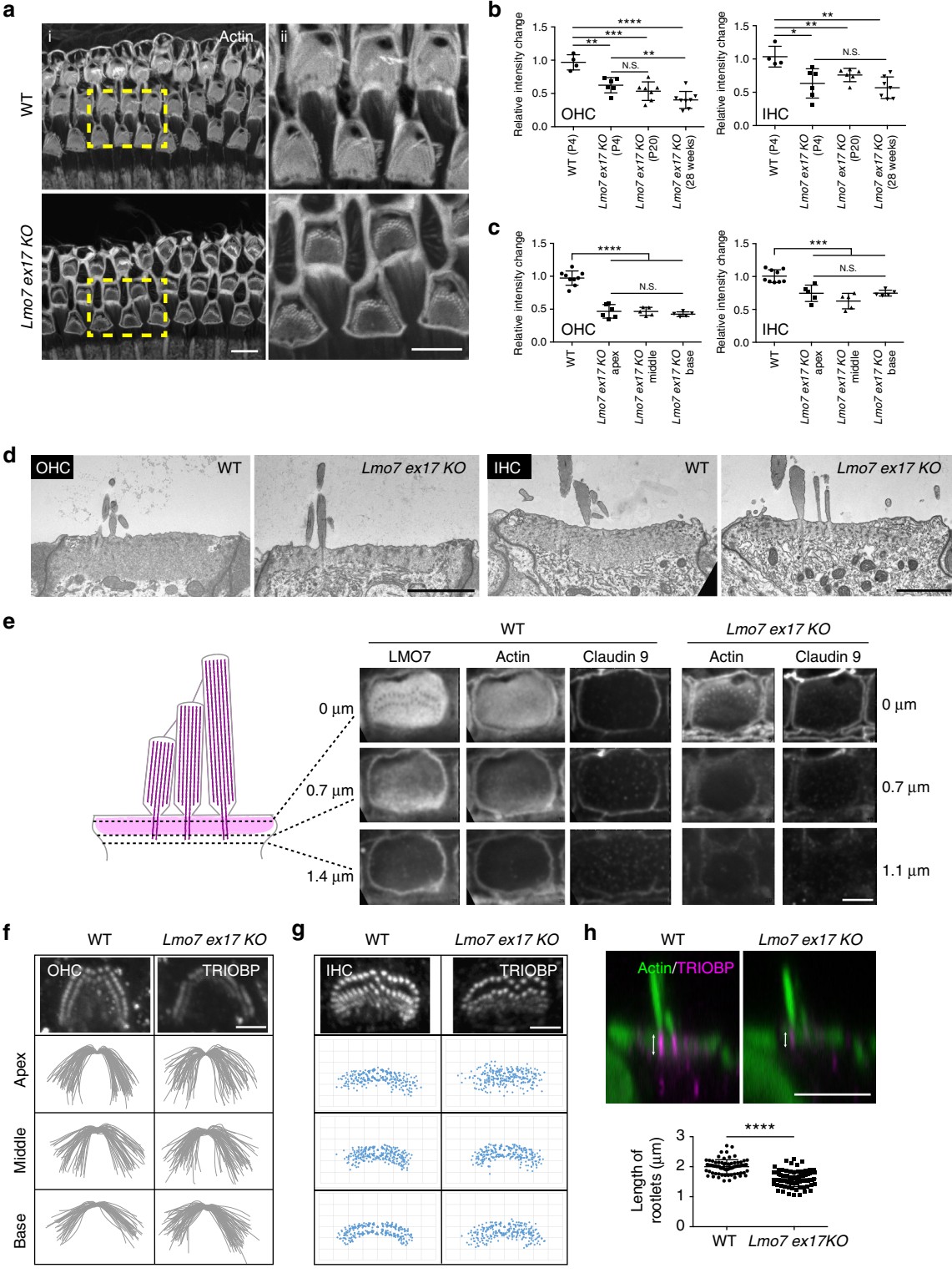

transfected with a construct coding for LMO7 tagged with GFP at the C-terminus. As illustrated in Fig. 5b, c (first columns), COS-7 cells transfected with WT LMO7-GFP exhibited a highly distinct F-actin organization, developing fibrillar and punctate condensations of F-actin that were mostly co-localized with LMO7-GFP. Such F-actin organization was not observed in untransfected COS-7 cells (Fig. 5b, c, first columns). To characterize the protein domains responsible for the F-actin re-organizing effect, we generated LMO7-GFP constructs in which the CH, LIM,

PDZ or the entire N-terminal third of LMO7 (LMO7-Δ1-353) was deleted (Fig. 5a). When overexpressed in COS-7 cells, LMO7-ΔCH and LMO7-ΔPDZ retained the F-actin-condensing activity (Fig. 5b). In contrast, LMO7-ΔLIM and LMO7-Δ1-353, while still co-localizing with F-actin, lost the ability to organize and condense F-actin (Fig. 5b). We suggest that both the LIM domain and a portion of the N-terminal domain bind F-actin and crosslink F-actin into dense networks. The putative actin-binding domains are indicated in Fig. 5a.

**Fig. 4** *Lmo7 KO* mice have defects in the cuticular plate. **a** (i) Reduced phalloidin staining in the cuticular plates of *Lmo7 exon17 KO* hair cells at P20. (ii) Enlarged view of boxed-in region. **b** Quantification of phalloidin reactivity in the cuticular plates of OHCs and IHCs in WT and *Lmo7 KO* mice, at different ages. The relative intensity changes are the ratio of phalloidin intensity of mutant and WT cuticular plates, at matched ages ($n = 15$–25 OHCs/mouse, $N = 8$, for P4 WT, $N = 6$ for P4 *Lmo7 exon17 KO* and WT, $N = 7$ for P20 *Lmo7 KO* and WT, $N = 8$ for 28 weeks *Lmo7 KO* and WT. $n = 5$–10 IHCs/mouse, $N = 8$ for P4 WT, $N = 6$ for P4 *Lmo7 KO* and WT, $N = 6$ for P20 *Lmo7 KO* and WT, $N = 7$ for 28 weeks *Lmo7 KO* and WT). **c** Quantification of phalloidin reactivity in the cuticular plate at different cochlear positions, at P30 ($n = 10$–20 OHCs/mouse, $N = 12$ for WT, $N = 6$ for *Lmo7 KO* and WT apex, $N = 5$ for *Lmo7 KO* and WT middle and base. $n = 5$–10 IHCs/mouse, $N = 12$ for WT, $N = 5$ for *Lmo7 KO* and WT apex, middle and base). **d** TEM analysis reveals thinner cuticular plates of OHCs and IHCs in *Lmo7 KO* compared to WT at 5 weeks. Scale bar, 2 µm. **e** Immunoreactivity of LMO7, actin and claudin 9 at three depth levels. **f** TRIOBP staining in OHC of WT and *Lmo7 KO* at p30, including line tracings of the first rootlets rows, indicating greater variability in *Lmo7 KO* mice ($n = 51$ for apex, $n = 56$ for middle and base, $N = 4$ for WT and *Lmo7 KO*). **g** TRIOBP staining in IHCs of WT and *Lmo7 KO* at p30. The rootlets were resolved well enough to allow scatter plot of individual rootlets. $n = 10$ IHCs, $N = 4$ for WT and *Lmo7 KO*. **h** Side-view of TRIOBP labeling in IHCs at P30, including quantification of rootlet length of the longest row of IHC stereocilia (as arrows shown in **h**) at P30. $n = 76$ stereocilia, $N = 4$ for WT and *Lmo7 KO*. Scale bar, 5 µm. Error bars indicate SD, ****$p$ value < 0.0001, ***$p$ value < 0.001, **$p$ value < 0.01, and *$p$ value < 0.05 in two-tailed unpaired Student's *t*-test

### Table 1 Co-Immunoprecipitation identifies LMO7-interacting proteins

| Protein ID | Spectral counts | | Fold change |
|---|---|---|---|
| | WT | LMO7 KO | |
| MYH14 | 41 | ND | Only in WT |
| GM17190 | 35 | ND | Only in WT |
| LMO7 | 97 | ND | Only in WT |
| KRT10 | 34 | ND | Only in WT |
| ATP5F1A | 21 | ND | Only in WT |
| KRT1 | 21 | 2 | 10.5 |
| ACTN4 | 20 | 2 | 10.0 |
| MYO6 | 29 | 5 | 5.8 |
| MYH9 | 100 | 19 | 5.3 |
| COCH | 78 | 24 | 3.3 |
| COL14A1 | 29 | 10 | 2.9 |
| SPTBN1 | 29 | 10 | 2.9 |
| CLTC | 29 | 12 | 2.4 |
| HIST1H4A | 38 | 16 | 2.4 |
| HIST1H1C | 27 | 12 | 2.3 |
| SPTAN1 | 28 | 14 | 2.0 |
| EEF1A1 | 28 | 15 | 1.9 |
| MPZ | 20 | 11 | 1.8 |
| SERPINH1 | 38 | 24 | 1.6 |
| HIST2H2AC | 23 | 15 | 1.5 |
| RPL7A | 21 | 15 | 1.4 |
| TUBA1B | 29 | 22 | 1.3 |
| IGHG3 | 23 | 18 | 1.3 |
| ACTB | 52 | 42 | 1.2 |
| TUBB5 | 40 | 37 | 1.1 |
| EWSR1 | 24 | 23 | 1.0 |
| HNRNPK | 29 | 28 | 1.0 |
| HNRNPUL1 | 20 | 20 | 1.0 |
| VIM | 32 | 34 | 0.9 |
| HSPA8 | 76 | 85 | 0.9 |
| HSPA9 | 53 | 60 | 0.9 |
| HNRNPH1 | 30 | 34 | 0.9 |
| DDX5 | 21 | 24 | 0.9 |
| HNRNPD | 21 | 24 | 0.9 |
| HNRNPU | 21 | 25 | 0.8 |
| HSPA5 | 103 | 135 | 0.8 |
| HNRNPA2B1 | 55 | 86 | 0.6 |
| DHX9 | 39 | 62 | 0.6 |
| HNRNPA1 | 37 | 62 | 0.6 |
| FUS | 37 | 76 | 0.5 |

List of 40 most abundant proteins in the WT pulldown, sorted by fold-enrichment in the WT vs. the KO pull-down. Columns indicate protein identifiers, spectral counts, and fold change in WT compared to *Lmo7 KO* pulldown

We next sought to confirm and characterize the interaction between LMO7 and its co-immunoprecipitating proteins, by co-expression in COS-7 cells. We focused on proteins that were previously reported to be localized in the cuticular plate (spectrin) or the cell junctions (NMII and ACTN4). LMO7-GFP and ACTN4 showed near perfect co-localization, in agreement with previous studies demonstrating direct interaction between these two proteins[28]. LMO7-GFP and spectrins (αII-Spectrin-MycDDk and βII-Spectrin-MycDDk were co-transfected and co-detected) display proximal, but not perfect co-localization in the network. The interaction between LMO7-GFP and NMII is characterized by a mix of colocalization and mutually-exclusive localization, reminiscent of the alternating pattern of alpha-actinin and NMII described previously for hair cell junctions[47] (Fig. 5c). We also tested whether the expression and localization of NMII and spectrin was affected in *Lmo7 KO* mice. Spectrin localization in the cuticular plate was less intense and uniform in the *Lmo7 KO* mice, but the pattern of NMII at the cellular junctions was not affected (Supplementary Fig. 2b). In summary, we conclude that LMO7 organizes the F-actin in the cuticular plate and the junctions into a dense network, by crosslinking F-actin, and serving as a scaffold for the complex interaction network between F-actin, spectrins, actinins, and NMII. This proposed role of LMO7 is consistent with the defects observed in the cuticular plate and the junctions of *Lmo7 KO* mice (Fig. 4).

**LMO7 deficiency affects cochlear tuning and sensitivity**. We reasoned that the reduced F-actin density and aberrant rootlet organization in LMO7-deficient cuticular plates might reduce its stiffness. Such an increase in compliance might impair the coupling of reticular lamina vibration and hair bundle deflection, potentially affecting the overall vibrations of the cochlear partitions. We tested the vibratory movements of cochlear partitions in vivo, using volumetric optical coherence tomography and vibrometry (VOCTV)[48]. The exquisite sensitivity of VOCTV enables the detection of subtle differences well before a hearing threshold shift is evident. Therefore, to isolate the effect of cuticular plate defects on cochlear vibrations from the effects of age in general, and from the *Cdh23*[ahl] allele responsible for age-related hearing loss in the C57Bl/6J strain, we performed VOCTV experiments at P30 on *Lmo7 exon 17* mice that were backcrossed to the CBA/J background (confirmed by 5307 SNP analysis, Dartmouse.org). Figure 6a, c show representative cross-sectional images of intact cochleae in live WT and *Lmo7 exon17 KO* mice obtained with VOCTV. Locations on the basilar membrane (BM), reticular lamina (RL), and tectorial membrane (TM) in the apical cochlear turn were selected for vibration measurements. Sound-evoked displacement magnitudes

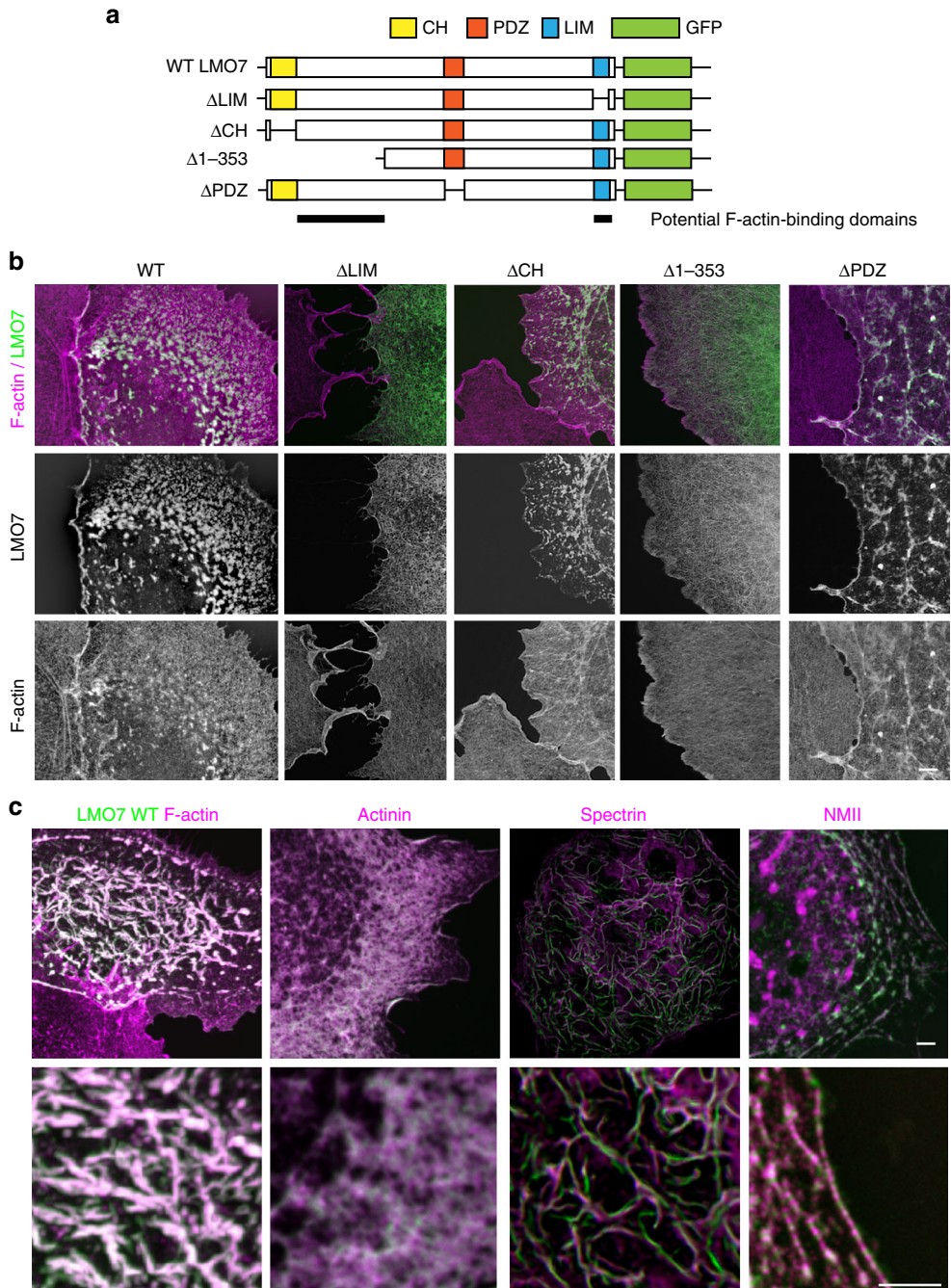

**Fig. 5** LMO7 re-organizes the F-actin network. **a** Schematic illustration of functional domains in LMO7 and corresponding domain deletion constructs (ΔPDZ, ΔLIM, ΔCH and Δ1-353aa). **b** Confocal imaging analysis of LMO7-WT, ΔPDZ, ΔLIM, ΔCH and Δ1-353aa fused to GFP, overexpressed in COS-7 cells. ΔPDZ and ΔCH constructs re-organize F-actin in a manner similar to full-length LMO7. ΔLIM and Δ1-353aa constructs remain co-localized with F-actin, but lose F-actin condensing capacity. **c** Confocal imaging analysis of LMO7-GFP co-transfected with spectrin, actinin or NMII in COS-7 cells. Scale bars, 2.5 μm

and phases measured from the BM, RL, and TM in WT and *Lmo7 exon17 KO* mice are shown in Fig. 6. Stimuli were swept from 2–14 kHz and varied in level from 10–80 dB SPL. Interestingly, the characteristic frequency (CF), which is defined as the frequency evoking the largest displacements at a low stimulus level (20 dB SPL), was slightly lower in the *Lmo7 exon17 KO* mouse (~10 kHz) than in the WT mouse (~11 kHz) for the TM, as well as the BM and RL (Fig. 6b, d). A reduction in CF is consistent with an element of the organ of Corti becoming more compliant, thus might be a direct consequence of the predicted reduction in stiffness in LMO7-

deficient cuticular plates. Vibratory sensitivity at the CF also tended to be lower in *Lmo7 exon17 KO* mice than in WT, though only significantly so for the RL (Fig. 6e, f). Taken together, the VOCTV measurements indicate that LMO7 deficiency effects changes in the frequency tuning and sensitivity of cochlear vibrations.

**LMO7 deficiency causes late-onset, progressive hearing loss.** Next, we examined the effects of the aforementioned defects of the cuticular plate and changes in cochlear vibrations on hearing function. We determined hearing thresholds in the *Lmo7 exon17*

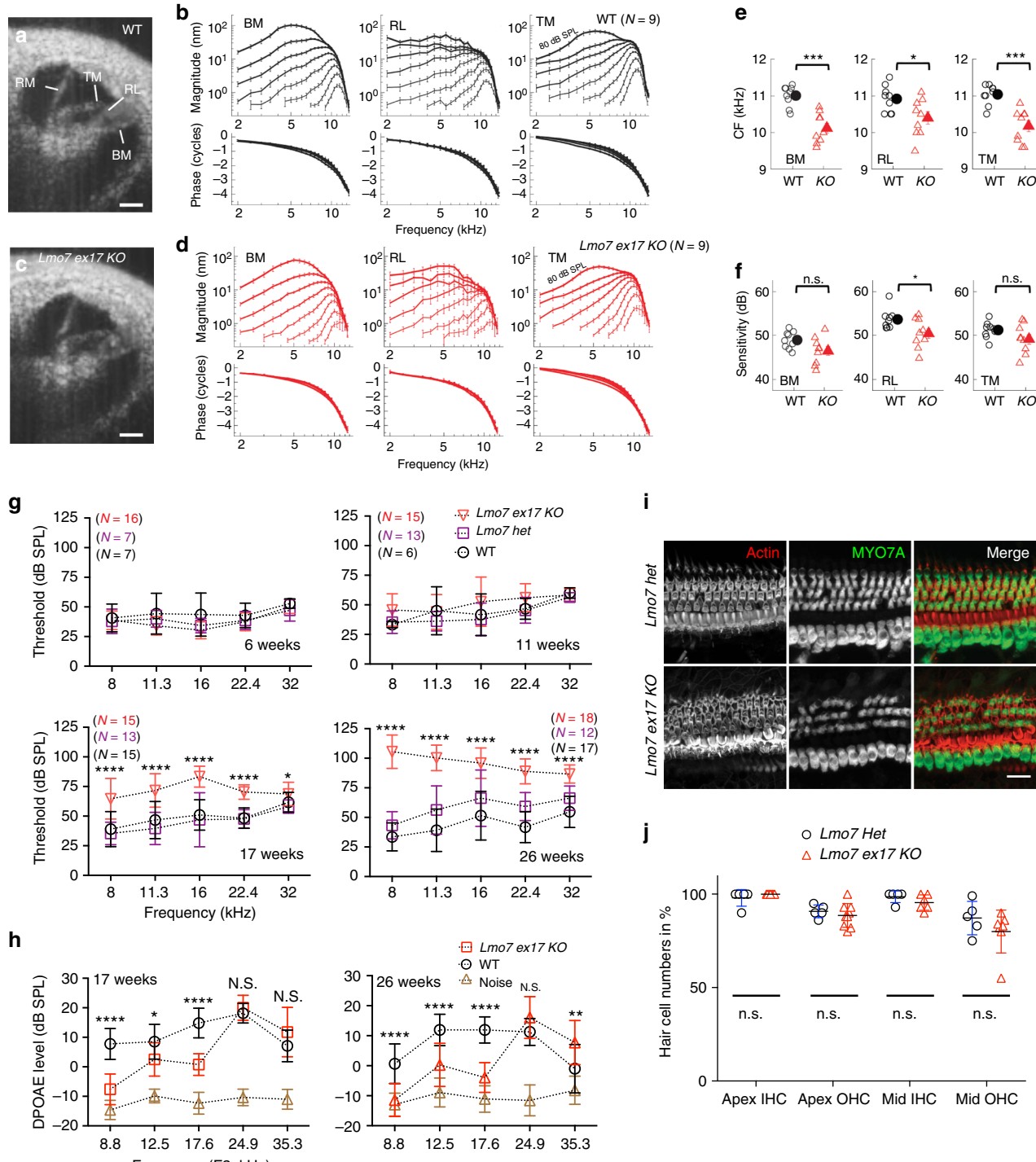

**Fig. 6** Abnormal cochlear vibrations and late-onset hearing loss in *Lmo7 KO* mice. **a**, **c** Representative cross-sectional images of intact cochleae in live WT **a** and *Lmo7 exon17 KO* **c** mice obtained with VOCTV. Vibrations were measured from specific points on the basilar membrane (BM), reticular lamina (RL), and tectorial membrane (TM) in the apical cochlear turn. Reissner's membrane (RM) is also indicated. Scale bars, 100 μm. **b**, **d** Average sound-evoked displacement magnitudes and phases measured from the BM, RL, and TM in WT **b** and *Lmo7 exon17 KO* **d** mice. Stimuli were swept from 2–14 kHz and varied in level from 10–80 dB SPL in 10 dB steps. TM displacement curve for the highest stimulus level is labeled for clarity. **e** Average CFs for the BM, RL, and TM were significantly lower in *Lmo7 KO* than in WT mice (individual data shown with open symbols). **f** Vibratory sensitivity at CF also tended to be lower in *Lmo7 KO* mice than in WT, though only significantly so for the RL. Sensitivity was defined as the ratio of the response magnitude for a 20 vs. an 80 dB SPL tone at CF, after dividing both displacements by the respective stimulus pressure. Error bars indicate SEM. *$p < 0.05$, ***$p < 0.0005$, n.s. = $p > 0.05$, by unpaired *t*-test. **g** ABR analysis demonstrates *Lmo7 exon17 KO* mice exhibit late onset, progressive hearing loss. **h** DPOAE output differs significantly between *Lmo7 exon17 KO* and WT littermates at 17 and 26 weeks. At 17 weeks: $N = 12$ for WT, $N = 13$ for *Lmo7 exon17 KO*. At 26 weeks: $N = 12$ for WT, $N = 14$ for *Lmo7 exon17 KO*. Error bars indicate SD, ****$p$ value $< 0.0001$, **$p$ value $< 0.01$, and *$p$ value $< 0.05$ (ANOVA, Tukey post-hoc test). **i**, **j** The hair cell numbers were not significantly affected in mutant mice (*Lmo7 exon17 het* control: $N = 5$, for *Lmo7 exon17 KO*: $N = 6$). Scale bar, 10 μm

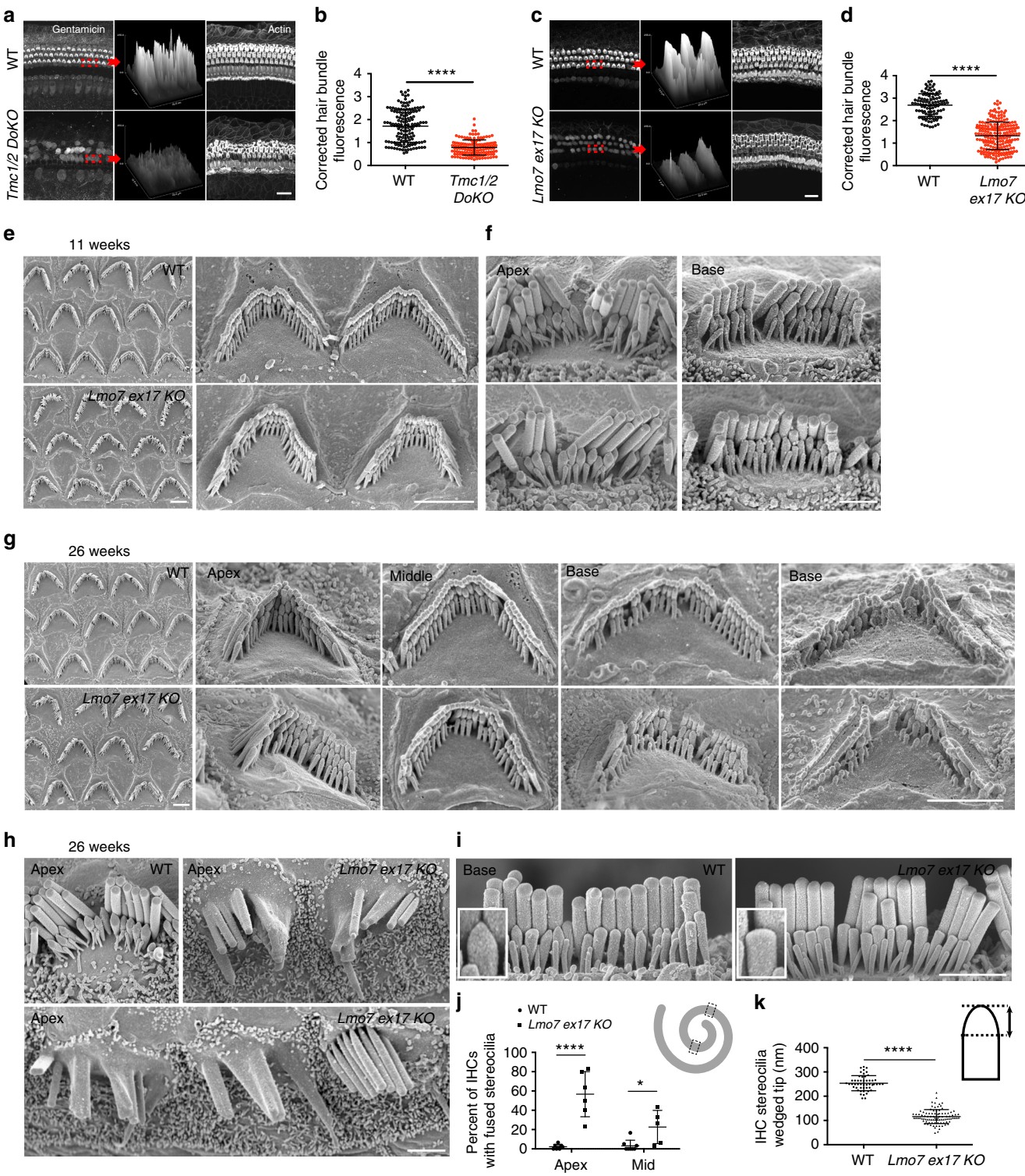

*KO* mice, using auditory brainstem response (ABR) measurements. Despite the fact that abnormalities in the cuticular plate are detectable in early postnatal animals and cochlear vibrations are altered at P30, hearing thresholds were comparable to WT controls until 11 weeks of age (Fig. 6g). However, ABR thresholds became significantly elevated by 17 weeks, and progressively worsened. By 26 weeks of age, *Lmo7 exon17 KO* mice developed near profound hearing loss (Fig. 6g). Interestingly, while threshold differences were significant at all frequencies tested at 26 weeks of age, the lower frequencies were most affected. The

hearing of *Lmo7 ex12 KO* mice were not affected until 26 weeks, while *Lmo7 gene trap* mice did not develop hearing loss at all (Supplementary Fig. 3). We also assessed OHC function using distortion product otoacoustic emissions (DPOAEs). At 17 weeks, a significant reduction in DPOAE levels was evident in the *Lmo7 exon17 KO* in mid- to low-frequency DPOAE output levels. DPOAE output was further reduced at 26 weeks of age (Fig. 6h).

To test whether hearing loss in *Lmo7 exon17 KO* mice is caused by loss of hair cells, we counted hair cell numbers in mice after their terminal ABRs at 26 weeks. MYO7A immunoreactivity

**Fig. 7** Reduced hair cell function and stereocilia defects in *Lmo7 KO* mice. **a** Validation of gentamicin uptake assay in *Tmc1/2 DOKO* mice at P28. Scale bar, 10 μm. **b** Quantitation of corrected hair bundle gentamicin uptake in WT and *Tmc1/2 DOKO* mice, $n = 148$ OHCs, $N = 3$ for WT, $n = 144$ OHCs, $N = 4$ for *Tmc1/2 DOKO*. **c** Gentamicin immunoreactivity, especially in the hair bundles, is significantly reduced in *Lmo7 exon17 KO* mice at 28 weeks, compared to WT controls. Scale bar, 10 μm. **d** Quantification of corrected hair bundle gentamicin uptake in WT and *Lmo7 exon17 KO* mice. $n = 104$ OHCs, $N = 3$ for WT, $n = 214$ OHCs, $N = 3$ for Lmo7 exon17 KO mice. Error bars indicate SD, ****$p$ value < 0.0001 in two-tailed unpaired Student's *t*-test. **e** Scanning electron micrographs (SEMs) of outer hair cells and **f** inner hair cells show no overt abnormalities at 11 weeks of age. **g** SEMs of outer hair cells show comparable hair bundle morphology in WT and *Lmo7 exon17 KO* mice at 26 weeks of age. Both genotypes show degeneration of basal hair cell stereocilia, which is expected in the BL6 background. **h** Inner hair cells in the apical region of *Lmo7 exon17 KO* have high incidence of fused stereocilia. Fusions were not detected in WT controls. **i** In basal hair cells of *Lmo7 exon17 KO* mice, stereocilia tenting was significantly reduced compared to WT controls, indicating reduced mechanotransduction activity. **j** Quantification of percent of IHCs with fused stereocilia within the regions labeled with dotted boxes (upper right cartoon), ($n = 7$ cochleae for WT apex, $n = 6$ cochleae for *Lmo7 exon17 KO* apex, $n = 8$ cochleae for WT middle, $n = 5$ cochleae for *Lmo7 exon17 KO* middle, $N = 4$ for both groups). **k** Heights of the basal IHC stereocilia wedged tips (upper right cartoon) in the second row of the hair bundle ($n = 50$ stereocilia from nine WT cells, $n = 92$ stereocilia from 13 *Lmo7 exon17 KO* cells, $N = 4$ for both groups). Error bars indicate SD, ****$p$ value < 0.0001 and *$p$ value < 0.05 in two-tailed unpaired Student's *t*-test

demonstrated that hair cell numbers were comparable between control and mutant cochleae (Fig. 6i, j), even in the apical regions corresponding to the frequencies that exhibit highly elevated thresholds in the ABR result of mutant mice. We conclude that *Lmo7 exon17 KO* mice develop late-onset, progressive hearing loss, especially at low frequencies. We further note that hearing loss is not caused by loss of hair cells.

**Reduced hair cell function in older *Lmo7 KO* mice.** Having established that hearing loss in *Lmo7 exon17 KO* mice is not caused by loss of hair cells, we investigated whether the mutant hair cells might be compromised in their mechanotransduction (MET) function. In the absence of an established method for assessing hair cell MET function in adult mice on a cellular level, we applied and quantified the uptake of the aminoglycoside gentamicin into hair cells in vivo as a surrogate for MET activity. Gentamicin was given intraperitoneally, in conjunction with the loop diuretic furosemide, which increases permeation of the blood-labyrinth-barrier[49,50]. As proof of principle, we first validated the usefulness of this technique as a readout for MET function, using MET-deficient *Tmc1/2 double KO (DOKO)* mice as negative controls. Mice were injected at P28, and were sacrificed 2 h later, prior to the onset of gentamicin-mediated ototoxicity. Cochleae were processed for staining with a gentamicin-specific antibody[51]. As described in Fig. 7a, b, hair cells in WT mice at P28 displayed robust uptake of gentamicin, especially in the stereocilia. Gentamicin immunoreactivity in *Tmc1/2 DOKO* hair cells was significantly reduced, and was not concentrated in the stereocilia as in WT controls (Fig. 7a, b). Some remaining gentamicin immunoreactivity was detected in the *Tmc1/2 DOKO* hair cells, likely because gentamicin enters hair cells through routes other than the MET channel, including endocytosis[52,53]. Having validated this method, we applied this method to test MET activity at 28 weeks, when *Lmo7 exon17 KO* mice exhibit near profound hearing loss. Compared to WT mice, *Lmo7 exon17 KO* mice showed a highly significant reduction in gentamicin immunoreactivity, in both inner and outer hair cells (Fig. 7c, d). We therefore conclude that the hearing loss in *Lmo7 exon17 KO* mice correlates with a reduction in mechanotransduction as inferred from gentamicin uptake.

**Stereocilia defects in older *Lmo7 KO* mice.** We next investigated the cause for the loss of MET activity. Abnormal cochlear tuning and reduced vibrational sensitivity were evident in young *Lmo7 exon17 KO* mice, and based on the age-progressive deterioration of the cuticular plate (Fig. 4b), these defects are expected to become worse with age. However, abnormal cochlear mechanics is unlikely to be the sole cause for the near profound hearing loss at 26 weeks of age. To identify direct correlates for loss of hair cell

function, we examined the ultrastructure of hair cells in *Lmo7 exon17 KO* mice, using scanning electron microscopy (SEM). At 11 weeks of age, the morphology of hair cells in *Lmo7 exon17 KO* is comparable to WT controls (Fig. 7e, f). The reduced F-actin density evident in phalloidin immunofluorescence thus does not cause overt changes in the surface morphology of the cuticular plate. Some hair bundles of outer hair cells of *Lmo7 exon17 KO* mice display some asymmetry of their characteristic W-shape (Fig. 7e), consistent with the aberrant TRIOBP staining at younger ages. At 26 weeks, the outer hair cell morphologies were comparable in appearance between WT and *Lmo7 exon17 KO* mice (Fig. 7g). In contrast, profound morphological defects, such as stereocilia fusion and loss, were detected in inner hair cells. These phenotypes were restricted to the apical cochlear regions (Fig. 7h, quantified in j), but more subtle changes in inner hair cell stereocilia were also evident in the other regions of the cochlea: the tenting of the second row of stereocilia, thought to be a correlate for functional tip links and active mechanotransduction (MET)[54], was reduced in inner hair cell stereocilia of mid-basal turns of *Lmo7 exon17 KO* mice (Fig. 7i, quantified in k). Taken together, we conclude that the loss of cuticular plate integrity, and subsequent abnormalities in cochlear mechanics and stereocilia degeneration give rise to late-onset progressive hearing loss in *Lmo7 exon17 KO* mice.

**Discussion**
While the impact of various genetic and environmental stressors on hair bundle integrity and hair cell function is well studied, the effect of cuticular plate defects on mammalian hearing function is not well understood. Our investigation into the role of LMO7 provides a unique avenue into a better understanding of the significance of the cuticular plate for the performance of the hair cell and the cochlea.

LMO7, expressed heterologously in COS-7 cells, induces condensation of F-actin into fibrillar and punctate patterns. Our findings thus suggest a role of LMO7 in organizing the F-actin in the cuticular plate and cell junctions into a dense network. Our study provided some insight into the mechanisms involved: known components of the cuticular plate (spectrin) and the junctions (NMII and alpha-actinin) were identified as potential direct or indirect interaction partners by Co-IP/MS. The F-actin modulatory effect of LMO7 was found to be mediated by two domains, the LIM domain and a domain located in the N-terminus of LMO7. We suggest that these two domains individually bind to actin, enabling LMO7 to crosslink F-actin filaments. LIM domains are known to bind F-actin[55], but the putative F-actin-binding domain in the N-terminus has not been characterized previously. Interestingly, the CH domain, known for its F-actin binding ability, was dispensable for the F-actin

organizing capacity of LMO7. This is consistent with our finding that the *Lmo7 gene trap* mouse line, which does not express the CH-harboring long isoform, exhibits WT-like F-actin density in the cuticular plate and retains normal hearing function (Supplementary Fig. 1b).

While not as severely affected as the cuticular plate, the cell junctions are also altered in *Lmo7 KO mice*: the F-actin and claudin 9 localization extends less deeply into the cell body, suggesting a mild atrophy of both the adherens and tight junctions. We suggest that LMO7 is a component of the specialized hair cell junctions, which were previously reported to contain features of both tight and adherens junctions[56].

The primary effect resulting from LMO7 deficiency is a reduction of F-actin density, thinning and abnormal rootlet organization in the cuticular plate. A previous study found that treating organ of Corti with the F-actin-disrupting toxin latrunculin A caused a 69% reduction in phalloidin reactivity at the surface of outer hair cells, along with an equivalent (64%) reduction in Young's modulus. The study concluded that the surface stiffness of outer hair cells was largely determined by F-actin[57]. Considering the ~50% reduction in phalloidin reactivity in the cuticular plates of *Lmo7 exon17 KO* mice, a corresponding reduction in cuticular plate stiffness might be expected. The cuticular plate and the hair bundle couple the outer hair cell's somatic motility to the overlying tectorial membrane. It is thus not surprising that the introduction of additional compliance into this crucial mechanical joint affects cochlear mechanics, as shown in the aberrant tuning and reduced sensitivity of cochlear vibrations in *Lmo7 exon17 KO* mice. These vibrational phenotypes are subtle and do not manifest in measurable hearing threshold changes in young animals, but progressive degradation of the cuticular plates ultimately affects DPOAE output and auditory brainstem responses starting at around 17 weeks of age. We suggest that the reduction in gentamicin uptake in 26-week-old mutant outer hair cells is also a consequence of reduced cochlear vibrations leading to decreased hair cell MET. While it is possible that the diminished uptake of gentamicin is caused by reduced endocytosis in the *Lmo7 exon17 KO* mice, the loss of stereocilia tenting, as evident in the SEM analysis, provides further corroboration of our hypothesis that the reduced gentamicin uptake reflects compromised MET activity.

In addition to the direct and early effects on the cytoskeletal integrity of the cuticular plate, LMO7 deficiency causes a late-onset degeneration of the inner hair cell stereocilia. In the apical and middle regions of 26-weeks-old mutant mice, many stereocilia were fused, and, in the base, the transducing stereocilia of the second row displayed significantly less tenting, indicative of reduced mechanotransduction. In apical hair cells harboring long stereocilia, the deterioration of cuticular plate rigidity and lack of rootlet organization could allow stereocilia to come in contact with one another and fuse. This hypothesis is consistent with the previously reported finding that mice deficient in the rootlet protein TRIOBP also show fusion of stereocilia[46]. The considerably shorter stereocilia in basal hair cells might be less prone to stereocilia fusion, restricting the fusion phenotype to the cochlear apex, but a more compliant cuticular plate might generate excessive strain on stereocilia tips along the whole length of the cochlea. Tip links could be broken by the strain, reducing tenting of stereocilia tips and ultimately causing MET dysfunction. We propose that both manifestations of stereocilia degeneration, stereocilia fusions and loss of tenting, contribute to the profound hearing loss in older mice, especially affecting low frequencies.

It was proposed that LMO7 is a nuclear shuttling protein[36,58], with the potential to relay information about the tensional state of the epithelium into transcriptional activity in the nucleus. In our analysis, neither antibody-based nor direct localization of tagged endogenous LMO7 using the split-GFP approach provided any convincing localization of LMO7 in the nucleus. At least in hair cells, it is thus unlikely that LMO7 plays a role in nuclear transcriptional activity.

Most genetic or environmental ototoxic stressors primarily impact high-frequency hearing, with the low frequency hearing affected to a lesser degree and with a delayed onset. Accordingly, most genetic models for hearing loss focus on high frequency hearing loss. The *Lmo7 exon17 KO* mouse is a rare experimental model for low frequency hearing loss, providing an opportunity to study the molecular mechanisms underlying this understudied type of hearing loss.

## Methods

**Animal care and handling**. The protocol for care and use of animals was approved by the University of Virginia Animal Care and Use Committee. The University of Virginia is accredited by the American Association for the Accreditation of Laboratory Animal Care. C57BL/6J (Bl6) and CBA/J mice used in this study were ordered from Jackson Laboratory (ME, USA). All mouse experiments were performed using C57BL/6J or CBA/J (Cochlear vibration measurements) inbred mouse strains. Neonatal mouse pups [postnatal day 0 (P0)–P4] were killed by rapid decapitation, and mature mice were killed by CO2 asphyxiation followed by cervical dislocation. *Lmo7 gene trap* mice were reconstituted by IVF using frozen sperm obtained from Texas A&M (mouse ID: Lmo7Gt(IST10208D3)Tigm). *Tmc1/2 DOKO* mice were provided by Dr. Andrew Griffith. *Lmo7 exon17 KO* mice were bred ten generations to the C57BL/6J or CBA/J backgrounds.

**Plasmid constructs**. Mouse *Lmo7* and *Actn4* were amplified by PCR from mouse inner ear cDNA and cloned into pcDNA-DEST47 and pcEF-DEST51, respectively. All *Lmo7* deletion constructs were created by using Q5 Site-Directed Mutagenesis Kit (NEB). αII-Spectrin-MycDDk (RC226485) and βII-Spectrin-MycDDk (RC212868) were obtained from Origene. NMII constructs were previously described[47].

**Immunocytochemistry**. Tissues were fixed for 30 min in 3% paraformaldehyde. After blocking for 1 h with 1% bovine serum albumin, 3% normal donkey serum, and 0.2% saponin in PBS (blocking buffer), organs were incubated overnight at 4 °C with primary antibodies in blocking buffer. Organs were then washed 5 min with PBS and incubated with secondary antibodies (7.5 μg/ml Alexa 647, Alexa 488, Alexa 555- conjugated donkey anti-rabbit IgG, donkey anti-mouse IgG, donkey anti-goat IgG, Invitrogen,) and 0.25 μM phalloidin-Alexa 488 (Invitrogen) in blocking buffer for 1–3 h. Finally, organs were washed five times in PBS and mounted in Vectashield (Vector Laboratories). Samples were imaged using Zeiss LSM880 and Leica confocal microscopes.

Imaging for Figs 1f, 4e, and 5 was performed on TiE inverted fluorescence microscopes (Nikon Instruments) equipped with a Apo 100 × 1.49NA objective, a Yokogawa CSU21 spinning disc confocal unit, and an Andor DU-897 EMCCD camera. Image acquisition and analyses was managed through NIS-Elements software (Nikon Instruments). Post-acquisition 3D deconvolution was performed using the automatically selected algorithms and parameters built in the NIS-Elements software.

The following antibodies were used: rabbit anti-LMO7 (M-300, sc-98422; lot# unknown, discontinued, Santa Cruz Biotechnology), mouse anti-LMO7 (B-7, sc-376807, lot# A1315, Santa Cruz Biotechnology, replacement product of rabbit M-300 antibody), rabbit anti-LMO7 (Sigma Prestige antibody, lot# R09596, HPA020923). M-300 and B-7 were raised against the same epitope and in our immunohistochemistry experiments, yielded highly comparable results. Mouse anti-Spectrin alpha chain (MAB1622; lot# 2586521, MilliporeSigma), V5 Tag Antibody (R960-25, lot# 1900119, Invitrogen), rabbit anti-TRIOBP (16124-1-AP, lot# 7387, Proteintech Group), rabbit anti-MYO7A (111501, lot# 111501, Proteus BioSciences), mouse anti-Gentamicin (16102, lot# 111092–060603, QED Bioscience Inc.), rabbit anti-NMII (PRB-444P-100, lot# D04, Covance), anti-Claudin-9 (Custom-made rabbit polyclonal antibody PB209, PRGPRLGYSIPSRSGA). The Claudin-9 antibody was originally described and validated in Nunes et al.[56].

**Immunoblots**. The whole P4 inner ear was homogenized and protein were isolated using Nucleic Acid and Protein Purification kit (Macherey-Nagel). Half of an inner ear per well was loaded onto the 12% Bis-Tris SDS PAGE gel (Novex 4–12%, Invitrogen), transferred to PVDF membranes, and stained with India Ink. Blots were then blocked in blocking buffer (ECL prime blocking reagent; GE Healthcare) for 1 h and probed with primary antibodies (overnight at 4 °C. After three 5 min washings in TBS/0.1% Tween 20, blots were incubated with HRP-conjugated goat anti-rabbit secondary antibody (Cell Signaling Technology) for 1 h, and bands were visualized by ECL reagent (Pierce Biotechnology ECL Western blotting substrate

and GE Healthcare GE ECL prime Western blotting reagent). Chemiluminescence was detected using an ImageQuant LAS4000 mini imager (GE Healthcare).

**CRISPR/Cas-mediated generation of mouse models.** Cas9 endonuclease mRNA was generated using the plasmid MLM3613 (provided by Keith Joung's lab through Addgene) as a template. The target sequence was chosen using the CRISPR Design bioinformatics tool, developed by Feng Zhang's lab at the Massachusetts Institute of Technology (crispr.mit.edu). The targets sequences used were as follows: CCTG CGTCAGGTGCCTACG on Lmo7 exon 12, GTGATGGACGTCCGGCGTTA on Lmo7 exon 17, AGAGGATGGGTTCCGCATGT on Lmo7 exon 28, GGTTTACAT CACATGGCGGT on Lmo7 exon 31 for GFP11 KI) were cloned downstream of the T7 promoter in the pX330 vector (provided by Feng Zhang's lab through Addgene). IVT was performed using the MAXIscript T7 kit (Life Technologies) and RNA was purified using the MEGAclear kit (Life Technologies). The repair template (GAGCTGACATTCCCGTTTCTGAATCTGAGTGCTTTTCTCTTC

TCTCCTTTGCCTAATCAGCTGGACGGCCAACCCGCGACCACATGGTG CTGCATGAGTATGTCAACGCTGCGGGAATTACCGCCATGTGATGTAAAC CCCCATTCGAAAGCGCTGTTGCAGATAGAAGAGGAGGCGCTTGTCACTG ATGCAGAGCTA) for GFP11 KI was made by Integrated DNA Technologies (Ultramer, PAGE-purified).

The *Lmo7 KO* mice and *Lmo7 GFP11 KI* mice were generated according to the procedure published previously[59]. Briefly, fertilized eggs produced from B6SJLF1 (The Jackson Laboratory) mating were co-injected with Cas9 protein (PNA Bio, 50 ng/μl) and sgRNA (30 ng/μl) with or without GFP11 repair template (10 ng/ul). Two-cell stage embryos were implanted on the following day into the oviducts of pseudopregnant ICR female mice (Envigo). Pups were screened by PCR and founders identified by DNA sequencing of the amplicons for the presence of indels and/or repair. All mouse strains described in this study are available upon request.

**AAV vector and organotypic explant cultures.** The scCMV-GFP1-10-W3SL AAV targeting vector was constructed by ligating a CMV promoter, GFP1-10, and a minimal woodchuck post-regulatory element[60] (Genbank accession #KJ411915), to a plasmid containing the internal terminal repeats corresponding to a self-complementary AAV[61] described previously[62]. AAV2/Anc80-GFP1-10 viral particles were prepared by the Gene Transfer Vector Core at Schepens Eye Research Institute, Massachusetts Eye and Ear. Mouse cochleae were dissected in Hank's balanced salt solution (HBSS, Invitrogen, MA) containing 25 mM HEPES, pH 7.5. The organ of Corti was separated from the spiral lamina and the spiral ligament using fine forceps and attached to the bottom of sterile 35 mm Petri dishes (BD Falcon, NY), with the hair bundle side facing up. The dissection medium was then replaced by two exchanges with culture medium (complete high-glucose DMEM containing 1% FBS, supplemented with ampicillin and ciprofloxacin). Prior to experimental manipulation, explants were pre-cultured for 24 h, to allow acclimatization to the culture conditions[51]. The cochleae were transduced with AAV2/Anc80-GFP1-10 for 24 h (virus was diluted 1:200 in the culture medium). After replacement of culture medium, cultures were maintained for 5 days, then fixed, and immunostained. The GFP1-10 coding sequence was followed by a DNA sequence coding for the V5 affinity tag. V5 immunoreactivity hence serves as transfection control.

**Immunoprecipitation (IP) and mass spectrometry.** Forty WT or *Lmo7 KO* P6 cochleae were homogenized with a pellet pestle homogenizer in lysis buffer (50 mM Tris (pH8), 100 mM NaCl, 0.5% NP40, 1 mM EGTA, 1 mM EDTA, 1x Protease inhibitor cocktail (04-693-116-001, Roche), 1 mM PMSF, 50 mM NaF, 0.2 mM Na3VO4) on ice. After centrifugation, the lysate was subjected to IP with 2 μg of monoclonal LMO7 antibody (B-7, sc-376807; Santa Cruz Biotechnology) overnight at 4 °C. Immuno-complexes were recovered by incubation with 30 μl of Protein G (GE17-0618-01, Sigma) for 2 h at 4 °C. Beads were washed three times in lysis buffer. The sample was reduced with 10 mM DTT in 0.1 M ammonium bicarbonate, then alkylated with 50 mM iodoacetamide in 0.1 M ammonium bicarbonate (both room temperature for 30 min). The sample was then digested overnight at 37 °C with 1 μg trypsin in 50 mM ammonium bicarbonate. Finally, the sample was acidified with acetic acid to stop digestion and pelleted by centrifugation. The solution was evaporated to 15 μL for MS analysis. The LC-MS system consisted of a Thermo Electron Q Exactive HFX mass spectrometer system with an Easy Spray ion source connected to a Thermo 75 μm × 15 cm C18 Easy Spray column. One micrgram of the extract was injected and the peptides eluted from the column by an acetonitrile/0.1 M formic acid gradient at a flow rate of 0.3 μL/min over 1.0 h. The nanospray ion source was operated at 1.9 kV. The digest was analyzed using the rapid switching capability of the instrument acquiring a full scan mass spectrum to determine peptide molecular weights followed by product ion spectra (10 HCD) to determine amino acid sequence in sequential scans. The data were analyzed by database searching using the Sequest search algorithm against Uniprot Mouse.

**Cell culture.** COS-7 cells were cultured in high-glucose DMEM (Life Technologies) supplemented with L-Glutamine, 1 mm sodium pyruvate (Life Technologies), 100 U/ml penicillin, and 100 μg/ml streptomycin (Life Technologies), and 10%

FBS (Life Technologies). Cells were maintained at 37 °C, 5% CO2. Plasmids were transfected into COS-7 cells with lipofectamine 3000 (invitrogen).

**Auditory brainstem response.** ABRs of adult WT, heterozygous and homozygous *Lmo7 exon 17 KO, Lmo7 exon 12 KO, Lmo7 gene trap KO* mice were recorded from ages 6 to 26 weeks. Mice were anesthetized with a single intraperitoneal injection of 100 mg/kg ketamine hydrochloride (Fort Dodge Animal Health) and 10 mg/kg xylazine hydrochloride (Lloyd Laboratories). All ABRs were performed in a sound-attenuating booth (Med-Associates), and mice were kept on a Deltaphase iso-thermal heating pad (Braintree Scientific) to maintain body temperature. ABR recording equipment was purchased from Intelligent Hearing Systems. Recordings were captured by subdermal needle electrodes (FE-7; Grass Technologies). The noninverting electrode was placed at the vertex of the midline, the inverting electrode over the mastoid of the right ear, and the ground electrode on the upper thigh. Stimulus tones (pure tones) were presented at a rate of 21.1/s through a high-frequency transducer (Intelligent Hearing Systems). Responses were filtered at 300–3000 Hz and threshold levels were determined from 1024 stimulus presentations at 8, 11.3, 16, 22.4, and 32 kHz. Stimulus intensity was decreased in 5–10 dB steps until a response waveform could no longer be identified. Stimulus intensity was then increased in 5 dB steps until a waveform could again be identified. If a waveform could not be identified at the maximum output of the transducer, a value of 5 dB was added to the maximum output as the threshold.

**Distortion product otoacoustic emissions.** DPOAE of adult WT and homozygous Lmo7 exon17 KO mice were recorded at 17 and 26 weeks. While under anesthesia for ABR testing, DPOAE were recorded using SmartOAE ver. 5.20 (Intelligent Hearing Systems). A range of pure tones from 8 to 32 kHz (16 sweeps) was used to obtain the DPOAE for right ear. DPOAE recordings were made for $f_2$ frequencies from 8.8 to 35.3 kHz using paradigm set as follows: $L_1 = 65$ dB, $L_2 = 55$ dB SPL, and $f_2/f_1 = 1.22$.

**Hair cell counts.** Mice were killed after the final ABRs at 26 weeks. Cochleae were dissected, openings were created at the base and apex of the cochlea and fixed in 4% paraformaldehyde (PFA) (RT-15720, Electron Microscopy Science, PA) for 24 h. After decalcification for 7 days in EDTA solution, apical and middle turns of the cochlear sensory epithelium were dissected. Hair cell counting was performed for the organ of Corti using MYO7A immunoreactivity as a marker of hair cell presence. After confocal microscopy, images were analyzed using ImageJ. Hair cells were counted from the apical (0.5–1 mm from apex tip, corresponding to the 6–8 kHz region), and mid turns (1.9–3.3 mm from the apex tip, corresponding to 12–24 kHz region) of the cochlea. Organ of Corti from at least 5 mice were analyzed for each experimental condition.

**Scanning electron microscopy.** Adult mice were killed via CO2 asphyxiation; animals were perfused intracardially, with 2.5% glutaraldehyde + 2% paraformaldehyde. The cochlea was dissected and a piece of bone was removed from the apex to create an opening. The fixative (2.5% glutaraldehyde, in 0.1 M caco-dylate buffer, with 3 mM CaCl2) was perfused through the round window and the apical opening. Cochleae were incubated in fixative overnight at 4 °C, then were decalcified in 4.13% EDTA, pH 7.3, for 10 days at room temperature. After the tectorial membrane was removed, the organ of Corti was dissected from the cochlea and was post-fixed in 1% OsO4, 0.1 M cacodylate buffer, and 3 mM CaCl2. The tissue was then processed according to the thiocarbohydrazide-OsO4 proto-col[63]. Cochleae were then dehydrated through a series of graded ethanol incubations, critical point dried, and mounted on stubs. After sputter coating with gold, they were imaged on a Zeiss Sigma VP HD field emission SEM using the secondary electron detector.

**Transmission electron microscopy.** Adult mice were perfused intracardially, with 2.5% glutaraldehyde + 2% paraformaldehyde. The cochleae were dissected and incubated in fixative overnight at 4 °C, then decalcified in 4.13% EDTA pH 7.3 in 2% paraformaldehyde, for 10 days at room temperature. The cochleae were post-fixed in 1% OsO4-1.5% ferricyanide in 0.1 M cacodylate buffer, then stained with 2% Uranyl Acetate. The samples were dehydrated in graded series of ethanol, propylene oxide, and 100% EPON. Samples were sectioned at 70 nm thickness onto 200 mesh copper grids and contrast-stained with lead citrate and 4% uranyl acetate. Imaging was conducted using a JEOL 1230 transmission electron microscope with an SIA 4Kx4K CCD camera.

**Cochlear vibration measurements.** VOCTV was used to image and measure vibrations from the intact mouse cochlea, as described in Lee et al.[48]. Mice (P28–P32) of either sex were anesthetized with ketamine (80–100 mg/kg) and xylazine (5–10 mg/kg), placed on a heating pad, and the skull was fixed to a custom head-holder with dental cement. A ventrolateral surgical approach was then used to access the left middle ear bulla, which was widely opened so that the otic capsule bone and middle ear ossicles could be visualized. VOCTV was performed using a custom-built system, which consisted of a broadband swept-source (MEMS-VCSEL, Thorlabs) with a 1310 nm center wavelength, 100 nm

bandwidth, and 200 kHz sweep rate, as well as a dual-balanced photodetector (WL-BPD600MA, Wieserlabs), and a high-speed digitizer (NI-5761, National Instruments). To obtain cross-sectional images of the apical cochlear turn, the source beam was scanned across the otic capsule bone using a 2D voice coil mirror attached to the dissecting microscope (Stemi-2000, Zeiss). Sound-evoked vibrations were then measured from specific points on the basilar membrane (BM), reticular lamina (RL) and tectorial membrane (TM), with 100 ms pure-tones presented via a speaker (MDR EX37B, Sony) positioned close to the eardrum. Displacement magnitudes and phases were obtained with stimulus frequency ranging from 2–14 kHz in 0.5 kHz steps and stimulus level varied from 10–80 dB SPL in 10 dB steps. After sacrificing the mouse via anesthetic overdose, displacements of the middle ear ossicular chain were measured. This was done so that the middle ear response phase could be subtracted from the cochlear vibration phase at each frequency, thus eliminating the influence of middle ear transmission delays. All displacement responses analyzed and shown here were required to have magnitudes falling at least three standard deviations above the mean of the measurement noise floor at surrounding frequencies.

**Gentamicin uptake assay.** Gentamicin sulfate (Fisher Scientific), was dissolved in 0.9% sterile saline to a concentration of 50 mg/ml. Adult mice received one subcutaneous injection of 100 mg/kg gentamicin. This was followed by a single intraperitoneal injection of furosemide (10 mg/ml; Hospira) at a dosage of 400 mg/kg 30 min later. The furosemide injection was used to enhance gentamicin uptake by increasing permeation of the blood-labyrinth-barrier. Mice were sacrificed 2 h later, prior to the onset of gentamicin-mediated ototoxicity. Cochleae were processed for staining with a gentamicin-specific antibody.

**Statistics.** For statistical analysis, GraphPad Prism (La Jolla, CA) was used. Two-way analysis of variance (ANOVA) was used to determine statistically significant differences in the ABR and DPOAE analyses. Significant differences in individual frequencies were determined by a Tukey post-hoc analysis. For two-tailed unpaired Student's $t$-test, $P$-values smaller than 0.05 were considered statistically significant. All n in statistical analyses refer to numbers of stereocilia, hair cells or cochlea regions, $N$ in statistical analyses refer to number animals. All error bars indicate standard deviation (SD) or standard error of the mean (SEM).

## Data availability

All relevant data are available from the authors upon request.

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

## Acknowledgements

We thank the Virginia Lion's Hearing Foundation (VLHF) for the generous financial support. This study was supported by NIH/NIDCD grants DC014254 (JBS), DC014450 and DC013774 (J.S.O), DC016211 (J.B.D) and in part by the NIH/NIDCD Intramural Research Program (Z01 DC000002) (BK).

## Author contributions

T.T.D., J.B.D., E.L.W., S.P.F., R.C., S.J.G., J.H., J.J.P., N.E.S., B.K. and J.B.S. performed the experiments. T.T.D., J.B.D., B.K. and J.B.S. analyzed the data. E.P.-R., W.X. and J.S.O. provided material and support. T.T.D. and J.B.S wrote the manuscript.

## Additional information

**Competing interests:** The authors declare no competing interests.

**Journal Peer Review Information**: *Nature Communications* thanks Ulrich Mueller and other anonymous reviewer(s) for their contribution to the peer review of this work. Peer reviewer reports are available.

