## [Peer Review File · Nature Communications]

Reviewers' comments:

Reviewer #1 (Remarks to the Author):

In this interesting manuscript, Du and colleagues provide insights into the molecular composition and function of the cuticular plate in sensory hair cells of the inner ear. Stereocilia of hair cells contain mechanically gated ion channels that are activated when stereocilia are deflected in response to sound. Stereocilia are anchored via rootlet filaments within the cuticular plate, which is thought to provide a rigid structural support at the base of stereocilia. The cuticular plate contains a network of actin filaments, but few other molecules are known that are components of the cuticular plate and regulate its formation, function and/or maintenance.

Du and colleagues now show convincingly that the LIM only proteins 7 (LMO7) is a component of the cuticular plate. Using genetically modified mice, the authors provide evidence that LMO7 is not essential for the formation of the cuticular plate but for its maintenance thus affecting the function of hair cells and leading to late onset hearing loss. The findings are novel and interesting and provide an entry point for further studies of this important structural compartment of hair cells.

I have several specific comments:

1. In the first paragraph of the results the authors state: " its localization was restricted to the cuticular plate and intercellular junctions (Fig. 1d-f)". Localization to intercellular junctions is not obvious from the figures. Maybe a panel with more detailed images of the junctions could be added.
2. The localization data using split-GFP are nice. Is hearing function affected in Lmo7-GFP11 KI mice. If so, the result is still interesting but it would be prudent to state this in the text. If the mice hear normally, this would be good to know too.
3. The phalloidin staining in Fig. 4 indicate structural defects in the cuticular plate. However, to clearly demonstrate these defects, transmission EM pictures would be important.
4. Is the integrity of the intercellular junctions maintained? This could be evaluated by staining for junctional proteins and/or TEM.
5. For the functional studies wild-type and mutants are compared but genetic background is important. The authors indicate that the mutants were backcrossed onto the CBA/J background but they do not state for how many generations. Usually 10 generations are considered suitable. If they used fewer generations, did they use as controls mice of a similar mixed background as mutant animals? This is especially important because defects in VOCTVs measurements are very small between controls and mutants.
6. The discussion is very speculative. The data in the manuscript do not really show that LMO7 mediates actin polymerization or that SRF is involved. It might be useful to reshape the discussion. Some speculation is fine, but maybe alternatives have then to be considered, or a better link to what we know about the cuticular plate. For example, could LMO7 be somehow linked to spectrin, which is also present in the cuticular plate?

Additional minor comments:

1. I think that the abstract could be improved. The authors state that hair cells have "two specialized organelles" but there are more than two (not just the hair bundle but cuticular plate are specialized;

the ER his highly unusual, IHCs have ribbon synapses, OHCs have a specialized lateral wall containing prestin, etc).

In addition, the authors state that “both structures (stereocilia and cuticular plate) have adapted to facilitate the remarkable sensitivity and speed of hair cell mechanotransduction”, but then they say “specifics about its function... are not known”. I think this could be formulated better. While the statements are likely fundamentally true they sound a little contradictory.

2. In the intro first sentence maybe: “...as evident in the sensory hair cells of the inner ear”..

3. in the second paragraph of the results: “...LMO7 is a novel hair cell-enriched protein”. I think “novel” is not the right term here. It is here for the first time shown to be in the cuticular plate, but is it not a novel protein.

Reviewer #2 (Remarks to the Author):

This manuscript by Du et al. reports on LMO7, a protein originally thought to be enriched in hair cell stereocilia but shown clearly here to be resident in the cell’s cuticular plate, which anchors stereocilia. Inactivation of the Lmo7 gene by CRISPR/Cas9 led to structural defects in the cuticular plate, e.g., reduced actin levels, and alterations in the patterning of stereocilia, presumably because LMO7 was needed for properly anchoring the stereocilia. The loss of Lmo7 also led to reduction in the extent and levels of TRIOBP, an important rootlet component.

The physiological effects of Lmo7 deletion were modest. Measurement of cochlear vibration at three key levels (basilar membrane, reticular lamina, tectorial membrane) showed very modest changes. Auditory brainstem response measurements demonstrated a slowly progressing hearing loss, intriguingly more prominent at lower frequencies. Hair bundles were disrupted, with effects being greater for inner hair cells than for outer hair cells.

The work in this manuscript appears to be of the highest quality, and presentation of the data is excellent. Several interesting and informative techniques were used (e.g., split GFP reconstitution; whole animal gentamicin labeling of hair cells to assess transduction). However, the manuscript is very short on mechanistic understanding. We clearly see that LMO7 is important for assembling a proper cuticular plate, but how? Otherwise the paper just documents (well, I should add) bits of information of LOM7, without the critical experiments that permit mechanistic understanding.

Major comments

1. LMO7 binding partners are not clear. The functional significance of LMO7 would be better understood if we knew what the protein binds to in the cuticular plate. A BioGRID search revealed possible binding to actin, nonmuscle myosin II, and CAPZ as possible components of the cuticular plate. How does LOM7 bind?

2. What exactly does LMO7 do in the cuticular plate? Similarly, it’s hard to assess the significance of the localization without knowing what the function LMO7 carries out in the cuticular plate. Yes, it appears to stabilize the cuticular plate, which otherwise disappears, but how?

Other comments

Impressively, I noticed no minor issues to be resolved.

Reviewer #3 (Remarks to the Author):

This manuscript from the laboratory of Jung-Bum Shin provides a nice description of a new gene that plays a role in hearing, Lmo7. Based on the results of a previous proteomic study, the authors sought to examine the role of Lmo7 in stereocilia. Lmo7 was, in fact, not found in stereocilia but in the adjacent cuticular plate. Generation of an Lmo7 mutant using CRISPR technology demonstrated late onset hearing loss. Additional experiments presented in the manuscript attempt to provide a biological basis for that hearing loss. These experiments are somewhat disjointed and incomplete, leaving this reviewer unsure that the study has actually been able to identify the biological basis for the hearing loss in these mice. In particular the following issues need to be addressed further:

Is Lmo7 expressed in other cells within the inner ear? The images are cropped tightly around the hair cells, making it impossible to tell if any other cochlear cells express the protein

Regarding the generation of Lmo7 mutant mice, some more details regarding the Lmo7 exon17 mutations would be very useful. What is the change in sequence? What is the predicted change in the protein? How does this relate with the loss of antibody labeling? What region of the normal Lmo protein is bound by the antibody? Based on the region of antibody recognition, is a loss of antibody labeling expected?

The VOCTV experiments are elegant and show a defect in RL that is intriguingly consistent with a defect in cuticular plates, but does not prove such a defect. Measurements of actual cuticular plate stiffness using AFM would be definitive. Also, how biologically relevant are these findings given that the ABR is normal at this age?

The use of gentamicin uptake as a proxy for defects in transduction seems logically flawed in the following way: Tmc1/2 cells are assumed to have no transduction channels but still take up Gentamicin, probably through endocytosis. Therefore, the decrease in uptake in the Lmo exon 17 cells could be a result of either a defect in transduction or a defect in endocytosis. Considering that the cuticular plate is thought to be the location of significant endocytosis in hair cells, this possibility should not be ignored. Further, was expression of Tmc1/2 examined in the mutants? Or, given the rounded morphology of the IHC bundles, were tip links examined either in SEM or by immunolocalization?

Author's response

We would like to thank the reviewer's for an opportunity to address the issues raised in their comments. We have conducted a significant body of additional experiments to address most concerns. We are happy to present a much improved manuscript, especially with regard to the mechanistic understanding of the role of LMO7 in the hair cell. In the revised manuscript, additions are indicated in **red font**, and deletions are indicated in ~~strikethrough~~. Please find below a point-by-point address of the critique points.

Reviewer #1 (Remarks to the Author):

In this interesting manuscript, Du and colleagues provide insights into the molecular composition and function of the cuticular plate in sensory hair cells of the inner ear. Stereocilia of hair cells contain mechanically gated ion channels that are activated when stereocilia are deflected in response to sound. Stereocilia are anchored via rootlet filaments within the cuticular plate, which is thought to provide a rigid structural support at the base of stereocilia. The cuticular plate contains a network of actin filaments, but few other molecules are known that are components of the cuticular plate and regulate its formation, function and/or maintenance.

Du and colleagues now show convincingly that the LIM only proteins 7 (LMO7) is a component of the cuticular plate. Using genetically modified mice, the authors provide evidence that LMO7 is not essential for the formation of the cuticular plate but for its maintenance thus affecting the function of hair cells and leading to late onset hearing loss. The findings are novel and interesting and provide an entry point for further studies of this important structural compartment of hair cells.

I have several specific comments:

1. In the first paragraph of the results the authors state: "its localization was restricted to the cuticular plate and intercellular junctions (Fig. 1d-f)". Localization to intercellular junctions is not obvious from the figures. Maybe a panel with more detailed images of the junctions could be added.

We have carried out new immunohistochemistry experiments to better visualize LMO7 localization at the junctions and its relation to other junctional proteins. We have also investigated the effect of LMO7 deficiency on the integrity of the intercellular junctions. In the manuscript, the corresponding new data on junctions can be found in Fig. 1F, Fig 4d,e.

2. The localization data using split-GFP are nice. Is hearing function affected in *Lmo7-GFP11* KI mice. If so, the result is still interesting but it would be prudent to state this in the text. If the mice hear normally, this would be good to know too.

For a meaningful assessment of hearing performance, we would have had to test ABR thresholds in 17 weeks old mice, which is the age when the *Lmo7* KO mice exhibit a threshold shift. This was not feasible in the time frame given for the revisions. Hearing performance of *Lmo7-GFP11* KI mice was evaluated up to 8 weeks of age and was found to be normal (not shown), which at least shows that LMO7-GFP11 does not exert a dominant-negative effect in hair cells. For the purpose this mouse line is used in the present study, which is to independently validate the localization of LMO7, we have confirmed that the expression level and subcellular localization of LMO7 in *Lmo7-GFP11* KI mice is indistinguishable from WT mice (new Fig. 2b_{iii}).

3. The phalloidin staining in Fig. 4 indicate structural defects in the cuticular plate. However, to clearly demonstrate these defects, transmission EM pictures would be important.

We agree. A TEM analysis is now included in the manuscript (Fig. 4d).

4. Is the integrity of the intercellular junctions maintained? This could be evaluated by staining for junctional proteins and/or TEM.

The revised manuscript adds significant amount of information about LMO7's role in the junctions. The TEM analysis in Fig. 4d indicates that the junctions are not grossly altered. However, double-labeling of LMO7 with the tight junction protein claudin-9 revealed a subtle atrophy of the hair cell junction (Fig. 4e). Finally, the Co-IP/MS data and the heterologous expression analyses added new insight into the interaction with junctional proteins such as actinin and NMII.

5. For the functional studies wild-type and mutants are compared but genetic background is important. The authors indicate that the mutants were backcrossed onto the CBA/J background but they do not state for how many generations. Usually 10 generations are considered suitable. If they used fewer generations, did they use as controls mice of a similar mixed background as mutant animals? This is especially important because defects in VOCTVs measurements are very small between controls and mutants.

We have backcrossed for seven generations, but on the first couple of backcrosses, have selected pups based on SNP analysis (semi-speed congenic). Furthermore, for the generation of mice used for the VOCTV experiment, we have confirmed their congenic CBA/J background by SNP analysis. As controls, we were therefore able to use non-littermate CBA/J WT mice.

6. The discussion is very speculative. The data in the manuscript do not really show that LMO7 mediates actin polymerization or that SRF is involved. It might be useful to reshape the discussion. Some speculation is fine, but maybe alternatives have then to be considered, or a better link to what we know about the cuticular plate. For example, could LMO7 be somehow linked to spectrin, which is also present in the cuticular plate?

Yes, we agree that the discussion of SRF was peripheral and have removed that part. The relation to spectrin has been addressed with new experiments (Co-IP/MS and Fig. 5).

Additional minor comments:

1. I think that the abstract could be improved. The authors state that hair cells have "two specialized organelles" but there are more than two (not just the hair bundle but cuticular plate are specialized; the ER his highly unusual, IHCs have ribbon synapses, OHCs have a specialized lateral wall containing prestin, etc).

In addition, the authors state that "both structures (stereocilia and cuticular plate) have adapted to facilitate the remarkable sensitivity and speed of hair cell mechanotransduction", but then they say "specifics about its function... are not known". I think this could be formulated better. While the statements are likely fundamentally true they sound a little contradictory.

Yes, we agree. We made appropriate changes to the abstract.

2. In the intro first sentence maybe: "...as evident in the sensory hair cells of the inner ear"..

We changed the sentence as suggested.

3. in the second paragraph of the results: "...LMO7 is a novel hair cell-enriched protein". I think "novel" is not the right term here. It is here for the first time shown to be in the cuticular plate, but is it not a novel protein.

Yes, we agree. We removed "novel".

Reviewer #2 (Remarks to the Author):

This manuscript by Du et al. reports on LMO7, a protein originally thought to be enriched in hair cell stereocilia but shown clearly here to be resident in the cell's cuticular plate, which anchors stereocilia. Inactivation of the *Lmo7* gene by CRISPR/Cas9 led to structural defects in the cuticular plate, e.g., reduced actin levels, and alterations in the patterning of stereocilia, presumably because LMO7 was needed for properly anchoring the stereocilia. The loss of *Lmo7* also led to reduction in the extent and levels of TRIOBP, an important rootlet component.

The physiological effects of *Lmo7* deletion were modest. Measurement of cochlear vibration at three key levels (basilar membrane, reticular lamina, tectorial membrane) showed very modest changes. Auditory brainstem response measurements demonstrated a slowly progressing hearing loss, intriguingly more prominent at lower frequencies. Hair bundles were disrupted, with effects being greater for inner hair cells than for outer hair cells.

The work in this manuscript appears to be of the highest quality, and presentation of the data is excellent. Several interesting and informative techniques were used (e.g., split GFP reconstitution; whole animal gentamicin labeling of hair cells to assess transduction). However, the manuscript is very short on mechanistic understanding. We clearly see that LMO7 is important for assembling a proper cuticular plate, but how? Otherwise the paper just documents (well, I should add) bits of information of LOM7, without the critical experiments that permit mechanistic understanding.

Major comments

1. LMO7 binding partners are not clear. The functional significance of LMO7 would be better understood if we knew what the protein binds to in the cuticular plate. A BioGRID search revealed possible binding to actin, nonmuscle myosin II, and CAPZ as possible components of the cuticular plate. How does LOM7 bind?

We agree that the lack of mechanistic data was a significant weakness of our original manuscript. Most of our efforts in the revision period has therefore focused on mechanistic experiments. In the revised manuscript, we are now able to present extensive data on the interaction partners and functional role of LMO7 in the cuticular plate and the junctions. In brief, we performed a Co-IP/MS experiment from organ of Corti lysates with an LMO7-specific antibody (with the *Lmo7* KO tissue as negative control) and

followed up on potential interaction partners using heterologous expression studies (mostly described in new Fig.5). We have also characterized the functional domains responsible for the F-actin organizing effect of LMO7. These new data significantly improved the overall manuscript, and we would like to thank the reviewer for facilitating this upgrade.

2. What exactly does LMO7 do in the cuticular plate? Similarly, it's hard to assess the significance of the localization without knowing what the function LMO7 carries out in the cuticular plate. Yes, it appears to stabilize the cuticular plate, which otherwise disappears, but how?

Much of the new data (Co-IP/MS, COS-7 expression analysis in Fig. 5) addresses this question. In a nutshell, our data suggest that LMO7 organizes the F-actin in the cuticular plate and the junctions into a dense network, by crosslinking F-actin, and serving as a scaffold for the complex interaction network between F-actin, spectrins, actinin and NMII.

Impressively, I noticed no minor issues to be resolved.

Reviewer #3 (Remarks to the Author):

This manuscript from the laboratory of Jung-Bum Shin provides a nice description of a new gene that plays a role in hearing, *Lmo7*. Based on the results of a previous proteomic study, the authors sought to examine the role of *Lmo7* in stereocilia. *Lmo7* was, in fact, not found in stereocilia but in the adjacent cuticular plate. Generation of an *Lmo7* mutant using CRISPR technology demonstrated late onset hearing loss. Additional experiments presented in the manuscript attempt to provide a biological basis for that hearing loss. These experiments are somewhat disjointed and incomplete, leaving this reviewer unsure that the study has actually been able to identify the biological basis for the hearing loss in these mice.

We hope that the data presented in the new Fig. 5, shedding light on the mechanisms by which LMO7 organizes the F-actin network, helps to alleviate the "disjointedness" of the original manuscript.

In particular the following issues need to be addressed further:

Is *Lmo7* expressed in other cells within the inner ear? The images are cropped tightly around the hair cells, making it impossible to tell if any other cochlear cells express the protein

We have now included a new Supplementary Figure 1a, featuring immunofluorescence localization of LMO7 in a large section of the organ of Corti (WT, at P4) that demonstrates highly specific expression in hair cells.

Regarding the generation of *Lmo7* mutant mice, some more details regarding the *Lmo7* exon17 mutations would be very useful. What is the change in sequence? What is the predicted change in the protein? How does this relate with the loss of antibody labeling? What region of the normal *Lmo* protein is bound by the antibody? Based on the region of antibody recognition, is a loss of antibody labeling expected?

We have now included the predicted, mutant LMO7 amino acid sequence in *Lmo7* exon 17 KO mice (**Fig. 3b**). The mutation leads to a premature stop codon five amino acid downstream of the mutation. The epitope recognized by the LMO7 antibody used exclusively for the original submission is located at the C-terminus of LMO7 and therefore is downstream of the premature stop codon of the mutant LMO7

variant (now indicated in Fig. 3a). The reviewer appears to ask whether it is possible that the *Lmo7 exon17 KO* mice still expresses a truncated protein that is just not recognized by the antibody. We agreed with this concern, and have carried out new immunohistochemistry and immunoblot experiments, with a new LMO7 antibody that recognizes an epitope upstream of exon 17 (rabbit anti-LMO7, cat. HPA020923, Sigma-Aldrich) (epitope indicated in Fig. 3a). Immunofluorescence and blot signals were abolished in *Lmo7 exon17 KO* tissue (Supplementary Fig. 1c), suggesting that the mutation causes either nonsense-mediated decay on the level of mRNA, or degradation of the truncated protein. Based on this data, we are confident that the *Lmo7 exon17 KO* is a *bona fide* loss-of-protein mouse model.

The VOCTV experiments are elegant and show a defect in RL that is intriguingly consistent with a defect in cuticular plates, but does not prove such a defect. Measurements of actual cuticular plate stiffness using AFM would be definitive.

We have considered the possibility of conducting atomic force microscopy experiments and consulted with various groups on and off-campus. The overwhelming feedback was that the lack of established methodologies and expected technical challenges in this regard makes this endeavor more suited for a dedicated biomechanics study in the future.

Also, how biologically relevant are these findings given that the ABR is normal at this age?

The significance of the VOCTV method lies in its extraordinary sensitivity. VOCTV can uncover defects in cochlear mechanics too subtle to manifest as ABR threshold shifts, therefore has the potential to indicate future defects in gross hearing function. With this in mind, we believe it is justified to perform VOCTV on mice with normal ABR thresholds.

The use of gentamicin uptake as a proxy for defects in transduction seems logically flawed in the following way: Tmc1/2 cells are assumed to have no transduction channels but still take up Gentamicin, probably through endocytosis. Therefore, the decrease in uptake in the *Lmo* exon 17 cells could be a result of either a defect in transduction or a defect in endocytosis. Considering that the cuticular plate is thought to be the location of significant endocytosis in hair cells, this possibility should not be ignored.

Considering corroborating data such as the loss of stereocilia tenting, we believe that reduced hair cell MET function is likely the cause for reduced gentamicin uptake. Nevertheless, we agree with the reviewer that alternative mechanisms such as reduced endocytosis might contribute to the reduction of gentamicin uptake. We now discuss this possibility in the Discussion section. Overall, the reviewer raises the very interesting possibility that a compromised cuticular plate and cell junction could affect vesicle trafficking and endo/exocytosis at the apical compartment. We hope that our study will encourage future studies in this regard.

Further, was expression of Tmc1/2 examined in the mutants? Or, given the rounded morphology of the IHC bundles, were tip links examined either in SEM or by immunolocalization?

The detection and localization of the very few molecules of TMC1/2 that make up the MET complex in adult mouse hair cells is exceedingly challenging if not impossible (past experience in the Shin and Kachar labs). Immunolocalization or SEM to visualize tip links in older mice is possible, but still very inconsistent. In our experience, the status of stereocilia tip tenting is a reasonable and robust morphological correlate of functional MET in adult hair cells. Exemplary reference: Schwander et

al. (2009) A mouse model for nonsyndromic deafness (DFNB12) links hearing loss to defects in tip links of mechanosensory hair cells. PNAS 106 (13) 5252-5257.

REVIEWERS' COMMENTS:

Reviewer #1 (Remarks to the Author):

The authors have done an excellent job in revising the manuscript. They have addressed all my concerns and even provide more mechanistic data regarding LOM7 function. This paper is a really important contribution to the field.

Reviewer #2 (Remarks to the Author):

The authors have done a stellar job in responding to my two substantive comments. They have provided substantial new data which illuminates the role of LMO7 much better.

Reviewer #3 (Remarks to the Author):

The authors have clearly made a significant effort to improve the manuscript based on the suggestions of the reviewers. The manuscript is much improved and now clearly demonstrates a role for LMO7 as an actin bundler or stabilizer in the cuticular plate. I have no additional comments or suggestions.

Response to reviewer's comments:

We would like to thank the reviewers for the reviews. The reviewers did not request any further experiments or data.

Reviewer #1 (Remarks to the Author):

The authors have done an excellent job in revising the manuscript. They have addressed all my concerns and even provide more mechanistic data regarding LOM7 function. This paper is a really important contribution to the field.

Reviewer #2 (Remarks to the Author):

The authors have done a stellar job in responding to my two substantive comments. They have provided substantial new data which illuminates the role of LMO7 much better.

Reviewer #3 (Remarks to the Author):

The authors have clearly made a significant effort to improve the manuscript based on the suggestions of the reviewers. The manuscript is much improved and now clearly demonstrates a role for LMO7 as an actin bundler or stabilizer in the cuticular plate. I have no additional comments or suggestions.